# Structure-guided functional studies of plasmid-encoded dihydrofolate reductases reveal a common mechanism of trimethoprim resistance in Gram-negative pathogens

Jolanta Krucinska [1], Michael N. Lombardo[1], Heidi Erlandsen [2], Alexavier Estrada [1], Debjani Si[1], Kishore Viswanathan[1] & Dennis L. Wright [1]✉

Two plasmid-encoded dihydrofolate reductase (DHFR) isoforms, DfrA1 and DfrA5, that give rise to high levels of resistance in Gram-negative bacteria were structurally and biochemically characterized to reveal the mechanism of TMP resistance and to support phylogenic groupings for drug development against antibiotic resistant pathogens. Preliminary screening of novel antifolates revealed related chemotypes that showed high levels of inhibitory potency against *Escherichia coli* chromosomal DHFR (EcDHFR), DfrA1, and DfrA5. Kinetics and biophysical analysis, coupled with crystal structures of trimethoprim bound to EcDHFR, DfrA1 and DfrA5, and two propargyl-linked antifolates (PLA) complexed with EcDHFR, DfrA1 and DfrA5, were determined to define structural features of the substrate binding pocket and guide synthesis of pan-DHFR inhibitors.

[1] Department of Pharmaceutical Sciences, University of Connecticut, 69N. Eagleville Rd., Storrs, CT 06269, USA. [2] Center for Open Research Resources & Equipment (COR2E), University of Connecticut, 91N. Eagleville Rd., Storrs, CT 06269, USA. ✉email: dennis.wright@uconn.edu

The rapid emergence of multidrug-resistant *Enterobacteriaceae* is a hallmark of the ongoing antimicrobial resistance crisis, highlighting our continued dependence on antibiotics and contributing to the healthcare burden and costs associated with infection management[1,2]. The discovery and development of effective antibiotics with novel mechanisms of action has been defined by a discovery void spanning 30 years and is complicated by the inability to find chemotypes able to penetrate the cell envelope of Gram-negative bacteria and reach their target[3]. An alternative approach is to expand existing classes of antibiotics by introducing structural modifications that overcome current resistance elements or to increase coverage to previously insensitive organisms. As demonstrated by the application of structure-guided design of carbapenems to overcome β-lactamase-mediated resistance[4], this strategy can be much more direct as it leverages a known chemotype with favorable properties to identify superior, next-generation agents. However, for many clinically useful antibiotics, much less is understood about the molecular basis of drug resistance that would accelerate the discovery of next-generation antimicrobials.

Targeting the folate pathway with the combination agent trimethoprim/sulfamethoxazole (TMP/SMX), has been one of the most successful treatment strategies for *Enterobacteriaceae* infections[5–7]. TMP is a selective inhibitor of bacterial dihydrofolate reductase (DHFR), which catalyzes the NADPH-dependent reduction of dihydrofolate (DHF) to tetrahydrofolate (THF), a necessary step for the recycling of reduced folate cofactors[8]. The metabolites are essential for the synthesis of deoxythymidine monophosphate (dTMP), purines, and amino acids such as glycine and methionine. DHFR inhibition is highly effective as an antibacterial approach since bacteria rely on this pathway as their sole source of dTMP; however, the effectiveness of trimethoprim has been diminished by drug resistance with rates reported as high as 60%[7,9].

Trimethoprim resistance in *Enterobacteriaceae* occurs almost exclusively through the acquisition of plasmid-associated *dfr* genes that encode intrinsically insensitive DHFR enzymes[10–12]. This acquisition creates functional redundancy in DHFR activity and converts the chromosomal DHFR into a non-essential gene. The two distinct families of DHFR enzymes that confer trimethoprim resistance, DfrA and DfrB, are encoded by the *dfrA* and *dfrB* genes, respectively. The DfrA proteins are homologous to chromosomal DHFR and constitute ~96% of all TMP resistance in Gram-negative bacteria[13]. Although more than 30 unique DfrA proteins have been identified; five predominant isoforms account for ~85% of all trimethoprim resistance in *E. coli*: DfrA1, DfrA5, DfrA7, DfrA12, and DfrA17[14,15]. Despite the

widespread resistance mediated by these proteins, little is known about the structural and mechanistic basis of trimethoprim insensitivity and how conserved those mechanisms are across this family of enzymes.

DHFR is a well-characterized enzyme, particularly with regard to the kinetics and dynamics of hydride transfer from NADPH to DHF in *E. coli* DHFR (*Ec*DHFR)[16,17]. Structurally, three functional loops play an important role in catalysis: the Met20 loop (residues 9–23), the F-G loop (residues 116–132), and the G-H loop (residues 142–149). Studies have shown these loops undergo major conformational changes, guiding *Ec*DHFR through five catalytic states[18,19]. Moreover, residue variations in these loops are known to modulate the rate of catalysis as well as trimethoprim resistance[20,21].

Herein, we describe comparative structural and mechanistic studies of two of the most clinically prevalent isoforms, DfrA1 and DfrA5, with *Ec*DHFR. Through phylogenic and sequence analysis, we identified two critical residue variations as a common structural element in trimethoprim-resistant DHFR, D27E and L28Q. The potential role of mutation to the acidic residue in TMP resistance has been considered in earlier work[22–25] Complementing biochemical and biophysical analysis defined how these substitutions mediate trimethoprim-resistance in DfrA1 and DfrA5 through effects on enzyme catalysis and ligand-cofactor cooperativity. These conclusions were supported by high-resolution structural data, paving the way for a design of next-generation broad-spectrum antifolates.

## Results and discussion

**Sequence analysis identifies D27E and L28Q substitutions as a common structural element in trimethoprim-resistant DHFRs.** Trimethoprim (TMP) competitively binds to the folate pocket, and resistance is driven by specific substitutions in the DHFR active site. Phylogenic studies of TMP-resistant DHFRs indicate that DfrA1 and DfrA5 are evolutionary related and fall within a large clade comprising nearly half of all DfrA proteins (Supplementary Fig. S1). Sequence alignment of the plasmid-borne isoforms identified substitutions at positions 27[EcDHFR] (Asp→Glu) and 28[EcDHFR] (Leu→Gln) as a common theme among TMP-resistant DHFR, and this D27E/L28Q substitution is conserved in 67% of DfrA proteins (Fig. 1). For the sake of clarity, all residue positions will be numbered relative to *Ec*DHFR unless otherwise noted.

In *Ec*DHFR, Asp27 forms a critical electrostatic interaction with the basic headgroup of dihydrofolate-competitive inhibitors[18,26]. Analysis of the crystal structures of DfrA1 (PDB ID 5ECX) suggested that the resulting increase in side chain length due to the

```
                    Met20 Loop
                 ┌──────────────┐
                    *      *   * *         *
EcDHFR   -MISLIAALA VDRVIGMENA MPWNLPADLA WFKRNTLDKP VIMGRHTWES IGRPLPGRKN   59
DfrA1    MKLSLMVAIS KNGVIGNGPD IPWSAKGEQL LFKAITYNQW LLVGRKTFES MG-ALPNRKY   59
DfrA5    MKVSLMAAKA KNGVIGCGPH IPWSAKGEQL LFKALTYNQW LLVGRKTFES MG-ALPNRKY   59

                                                              ┌
EcDHFR   IILSSQPGTD DR--VTWVKS VDEAIAACG- DVPEIMVIGG GRVYEQFLPK AQKLYLTHID   116
DfrA1    AVVTRSSFTS DNENVLIFPS IKDALTNLKK ITDHVIVSGG GEIYKSLIDQ VDTLHISTID   119
DfrA5    AVVTRSAWTA DNDNVIVFPS IEEAMYGLAE LTDHVIVSGG GEIYRETLPM ASTLHISTID   119

            F-G Loop                      G-H Loop
         ┌────────────┐              ┌──────────┐
EcDHFR   AEVEGDTHFP DYEPDDWESV FSEFHDADAQ NSHSYCFEIL ERR                   159
DfrA1    IEPEGDVYFP EIP-SNFRPV FTQDFA---- SNINYSYQIW QKG                   157
DfrA5    IEPEGDVFFP NIP-NTFEVV FEQHFS---- SNINYCYQIW QKG                   157
```

**Fig. 1 Sequence alignment of *E. coli* DHFR (EcDHFR) with DfrA1 and DfrA5.** Variations in the Met20 loop (blue), F-G loop (green), and G-H loop (red) are not only responsible for trimethoprim resistance, but also explain differences in steady-state kinetics. Notable active site variations (*) are directly involved in trimethoprim resistance.

D27E substitution would reposition ligand, reducing TMP affinity while preserving substrate binding[27,28]. In the case of L28Q substitution, Leu28 is proximal to hydrophobic regions of dihydrofolate and replacement by glutamine in DfrA proteins would likely alter substrate/inhibitor binding mode based on the same crystallographic evidence.

Residue variations at Asp27 and Leu28 are known to have effects on $Ec$DHFR catalysis. The D27E mutation was shown to increase ligand dissociation rates while maintaining the enzyme's ability to turnover substrate[29]. This in turn might suggest that competitive inhibitors would be impacted by the D27E substitution while the enzymatic activity would be preserved. Likewise, previous studies of DHFR variants demonstrated that mutations at $Ec$DHFR's Leu28 play an important role on enzyme efficiency and resistance to TMP[30]. In addition, Wanger et al.[31] found that the L28F mutation increased $k_{cat}$ through interactions with the substrate/product. We proceeded to introduce these substitutions in the chromosomal reductase to generate $Ec$DHFR mutants harboring D27E, L28Q, and D27E/L28Q substitutions as important comparators for our mechanistic studies on the DfrA1 and DfrA5 proteins.

## DfrA1, DfrA5, and $Ec$DHFR D27E/L28Q variants confer high levels of trimethoprim resistance.

To probe the effects of active site variations on DHFR inhibition, a panel of inhibitors including TMP, methotrexate (MTX), and iclaprim (a bacterial DHFR inhibitor in clinical development[32]) was assembled and tested in a spectroscopic assay that monitors the oxidation of NADPH. In addition, we conducted a preliminary screen of an in-house antifolate collection based upon the propargyl-linked antifolate (PLA) scaffold designed to target TMP-insensitive DHFR enzymes[33–38]. Similar to trimethoprim, PLAs bind to the DHF binding site via a diaminopyrimidine ring (A-ring). However, the extended spectrum of activity is mediated through a highly functionalized biphenyl system (B- and C-rings). While several PLAs demonstrated activity against all three enzymes, two promising compounds, UCP1223 and UCP1228 (Fig. 2) along with TMP, were selected for biochemical and structural studies (Tables 1–3).

DfrA1 and DfrA5 showed varying degrees of sensitivity to the clinically relevant DHFR inhibitors (Table 3). TMP displayed a 1000- to 3000-fold loss of activity against DfrA1 and DfrA5 compared to $Ec$DHFR, as did iclaprim (60–700-fold loss) and MTX (520–1700-fold loss). The weak activity of MTX was surprising and is likely affected by the same structural factors responsible for the increased $K_M$ of DHF. Pleasingly, UCP1223 and UCP1228 maintain potent inhibitory activity against both TMP-resistant enzymes with $K_i$ values ranging between 16 and 30 nM, only a 10–14-fold reduction relative to $Ec$DHFR, providing compelling evidence that it should be possible to develop pan-DHFR inhibitors.

The improved PLA activity against the $dfrA1$ and $dfrA5$ proteins arise from productive contacts made possible by the extended biphenyl moiety of the inhibitors. One consequence of this design is that the PLAs possess increased molecular weight and increased hydrophobicity relative to TMP. These physical properties can exert a strong impact on bacterial penetration, especially in Gram-negative pathogens.

The compounds were tested against a standard $E. coli$ strain (BW25113) and an isogenic mutant (JW0451) with an ACR-B knockout to assess the impact on both membrane permeability and efflux. Both UCP1223 and UCP1228 showed moderate activity against the BW25113 (20 and 10 µg/mL respectively) versus 0.312 µg/mL for TMP. However, the activity of the two PLA inhibitors was greatly increased to 1.25 µg/mL and 0.625 µg/mL

respectively in the ACR-B mutant, showing that efflux was a major factor limiting cellular accumulation, and that the PLA retain the ability to permeate the Gram-negative membrane. As expected, strong potentiation of the PLAs was observed by pairing with sulfamethoxazole, yielding MIC values of 0.625 µg/ml for both compounds. The effects of the PLA leads were also evaluated in the background of the dfrA1 and dfrA5 resistance element by transforming the $E.coli$ BL21(DE3) cell with the expression plasmids for these enzymes under the T7 promoter. As expected, a high level of expression of these DHFRs resulted in a substantial decrease in the susceptibility of the strains harboring DfrA1 and DfrA5 to both PLAs, with MIC values ranging from 10 to 20 µg/ml in the presence of SMX. Yet, this level of antimicrobial activity exerted by UCP1223 and UCP1228 is a clear enhancement over a lack of growth inhibition observed with TMP/SMX combination (MCIs of >20 µg/ml, Supplemental Table S3). Efforts to further optimize the PLAs to improve permeability and reduce the efflux liability are ongoing.

To determine if the mutants would recapitulate sensitivity trends seen with the plasmid-encoded enzymes, we extended the enzymatic activity assay to $Ec$DHFR D27E, L28Q, and D27E/L28Q variants (Table 3). The D27E substitution resulted in a moderate increase in $K_i$ over $Ec$DHFR for TMP, UCP1223 and UCP1228 (16-fold, 8-fold, and 4-fold, respectively) while comparable shifts in $K_i$ were also observed with L28Q mutation. The combinatorial mutant displayed 98-, 118-, and 69-fold loss in affinities, respectively, suggesting that these two single substitutions are important contributors to TMP insensitivity, but they must be present in combination to confer the high levels of resistance associated with the plasmid-encoded reductases.

## Steady-state kinetics link D27E and L28Q to changes in catalytic efficiency.

All enzymes displayed Michaelis–Menten kinetics with DfrA1 and DfrA5 exhibiting 3-fold and 8-fold increase in DHF $K_M$ and 2-fold and 3-fold increase in NADPH $K_M$ compared to $Ec$DHFR (Table 3). The diminished affinity for substrate and cofactor is offset by an increased rate of catalysis ($k_{cat}$), 2- and 5-fold for DfrA1 and DfrA5, respectively, indicating that these enzymes rely on compensatory mechanisms that result in catalytic efficiencies ($k_{cat}/K_M$) comparable to the chromosomal DHFR.

Similar trends were seen when the D27E or L28Q mutations were introduced in the chromosomal enzyme; a 1.5-fold (L28Q) and 5-fold (D27E) increase in DHF $K_M$ values. Negligible effects on NADPH affinity values stem from the fact that these residues are distal to the cofactor binding site. Interestingly, there was a corresponding increase in reactivity ($k_{cat}$) for D27E mutant, mimicking the compensatory effects observed in DfrA1 and DfrA5. The reduction in the Michaelis constant (DHF $K_M$) was 18-fold exaggerated in the D27E/L28Q mutant, relative to $Ec$DHFR. However, there was actually a 4-fold decrease in $k_{cat}$. The overall 70-fold change in catalytic efficiency for the D27E/L28Q variant is far greater than the sum of the respective effects of the individual substitutions and suggests that other subtle changes in the DfrA enzymes are required when these substitutions are present in combination.

Taken together, these results illustrate the constitutive D27E/L28Q substitutions govern ligand recognition in the folate active site through repositioning of the substrate such that the reaction is faster, presumably through increased product release. This conclusion aligns with what is known about DHFR's catalytic cycle as either hydride transfer or product release are the rate-limiting steps. Previous studies have shown that Asp27 is important for facilitating protonation of DHF and subsequent hydride transfer to generate THF[39]. Additionally, Leu28 mutants have been shown to increase THF dissociation[31]. Structurally,

**Fig. 2 Substrates and Inhibitors of DHFR. a** Dihydrofolate reductase (DHFR) converts dihydrofolate (DHF) to tetrahydrofolate (THF) by protonation form solvent and hydride transfer from NADPH. **b** The structures of clinically relevant DHFR inhibitors and propargyl-linked antifolates, UCP1223 and UCP1228.

**Table 1 Data collection and refinement statistics (molecular replacement).**

|  | EcDHFR:TMP | EcDHFR:NADPH:TMP | DfrA1:TMP | DfrA5:NADPH:TMP |
|---|---|---|---|---|
| Beam line | NSLS-II 17-ID-1 (AMX) | NSLS-II 17-ID-1 (AMX) | NSLS-II 17-ID-1 (AMX) | SSRL 14-1 |
| PDB ID | 7NAE | 7MYM | 7MYL | 7R6G |
| Space group | $P6_122$ | $C222_1$ | $P2_1$ | $P4_3$ |
| # monomers in ASU | 1 | 3 | 6 | 2 |
| Unit cell |  |  |  |  |
| $a, b, c$ (Å) | 65.54, 65.54, 216.08 | 65.47, 113.40, 216.58 | 55.07, 72.89, 125.12 | 99.53, 99.53, 42.82 |
| $\alpha, \beta, \gamma$ (°) | 90, 90, 120 | 90, 90, 90 | 90, 90.56, 90 | 90, 90, 90 |
| Resolution range (Å) | 56.83–2.35 (2.35–2.41) | 56.76–3.04 (3.25–3.04) | 63.06–2.15 (2.21–2.15) | 99.53–2.61 (2.81–2.61) |
| Completeness (%) | 99.8 (96.1) | 98.7 (98.1) | 98.6 (97.8) | 99.9 (100) |
| # unique reflections | 11,174 (823) | 15,680 (2797) | 53,224 (4327) | 13,057 (1579) |
| $I/\sigma$ | 20.3 (4.8) | 3.9 (1.9) | 7.6 (1.9) | 7.9 (2.0) |
| $R_{work}/R_{free}$ | 0.203/0.262 | 0.215/0.282 | 0.208/0.281 | 0.194/0.225 |
| # of atoms |  |  |  |  |
| Protein | 1267 | 3879 | 7434 | 2470 |
| Ligands | 46 | 227 | 126 | 126 |
| Solvent | 35 | 13 | 50 | 12 |
| Average B-factor |  |  |  |  |
| Protein | 61.73 | 55.14 | 31.48 | 45.0 |
| Ligands | 68.44 | 56.11 | 4.62 | 55.4 |
| Solvent | 55.91 | 30.55 | 17.39 | 35.6 |
| RMS (bonds) (Å) | 0.009 | 0.014 | 0.022 | 0.005 |
| RMS (angles) (°) | 1.62 | 1.82 | 1.92 | 1.38 |
| Ramachandran plot |  |  |  |  |
| Favored (%) | 95.5 | 94.7 | 96.0 | 97.7 |
| Allowed (%) | 2.6 | 4.7 | 3.9 | 6.4 |
| Outliers (%) | 1.9 | 0.6 | 0.1 | 0.7 |
| Rotamer outliers (%) | 8.8 | 7.2 | 6.1 | 6.4 |

Values in parentheses are for highest-resolution shell.

these residues are thus posed to impact inhibitor binding and facilitate ligand/cofactor interactions.

**Thermal unfolding data define NADPH as a major contributor to DHFR-ligand complex stabilization**. A label-free thermal scanning intrinsic fluorescence technique (Tycho NT.6, Nano-Temper Technologies) was utilized to calculate the inflection temperature ($T_i$) as a marker of protein stability. The melting curve for EcDHFR showed three unfolding transitions ranging from 50 to 62 °C, indicative of an equilibrium of transient folding intermediates with the native-like secondary structures (Supplemental Table S1 and Supplemental Fig. S2). These multiple unfolding events coincide with thermodynamic studies of EcDHFR using multiple spectroscopic probes[40]. For the L28Q and D27E/L28Q mutants, unfolding transitions were increased by almost 2 °C and 4 °C respectively, relative to EcDHFR, suggesting that the L28Q substitution may provide a stabilizing advantage. Meanwhile, the D27E substitution had only negligible effects on thermal stability, falling within the same inflection temperature as EcDHFR.

We then assessed the unfolding profile of EcDHFR, DfrA1 and DfrA5 as apo forms, binary complexes (protein preincubated with TMP or NADPH), and ternary complexes (protein premixed with both TMP and NADPH) (Table 4 and Supplemental Fig. S3). The range of $T_i$ values determined for EcDHFR, DfrA1 and DfrA5 apo forms (60.2 °C, 49.8 °C, and 66.3 °C, respectively) points to unique conformational variations between the enzymes. Using molar excess of TMP and NADPH to ensure complex formation, all three proteins display ligand-dependent stabilization, as is evident by a gradual increase in $T_i$ of the binary complexes culminating in high transition temperatures for the ternary systems. While the presence of either ligand during heat treatment had a stabilizing effect across all enzymes, the extent of a thermal shift caused by either TMP or NADPH showed a distinct pattern. Specifically, the EcDHFR:TMP complex displayed a stabilizing shift of 13.9 °C compared to the apo form. Conversely, thermal unfolding of the trimethoprim-binary complexes with DfrA1 and DfrA5 produced minimal shifts of 2.5 °C and 0.5 °C, respectively. NADPH contributed fewer stabilizing forces than TMP in EcDHFR, but more so in binary complexes with DfrA1 and DfrA5. This reversal of a response trend seen upon addition of TMP and NADPH in the TMP-resistant DHFRs supports the notion of some perturbations to the folate pocket leading to enhanced inhibitor dissociation as manifested by exceptionally high Ki values determined for TMP against DfrA1 ($K_i$: 1,332 nM) and DfrA5 ($K_i$: 394 nM).

Altogether, the EcDHFR ternary complex showed additive stabilizing effects of NADPH and TMP, whereas in the DfrA1 and DfrA5 systems NADPH had a clear advantage over TMP, underscoring that the DfrA:ligand formation is far more dependent on cofactor-inhibitor interactions. Under the same non-equilibrium conditions, UCP1223 and UCP1228 demonstrated similar effects on DHFR stability as TMP, increasing the overall thermal transition of all complexes (Supplemental Fig. S4). Interestingly, the UCP1223 had a distinctly stabilizing effect across all three enzymes, agreeing well with our crystal structures where a racemic mixture of this ligand is likely adopting different binding poses for an optimal configuration. In each case, addition of NADPH moved the reactivity of the binary complexes into highly stable conformers with an overall net shift of more than 21 °C. Most importantly, the cooperative effect of NADPH-inhibitor interactions on thermal stability is not specific or limited to TMP, as it is also present with PLAs. This data supports the idea that DHFR inhibition is strongly influenced by the relative positioning of the antifolate and NADPH cofactor, such that an

increase in cofactor:ligand:enzyme interactions strongly stabilize the ternary complex[41,42].

**Loss of NADPH-ligand cooperativity is a hallmark of trimethoprim resistance**. Based on our thermal stability data, the increase in structural integrity of EcDHFR through NADPH-induced changes was further enhanced by the binding of TMP. A less pronounced change in the thermal stability observed for DfrA1 and DfrA5 suggests that a reduction in NADPH-TMP cooperativity plays an important role in enzyme inhibition. Towards this end, we determined the equilibrium dissociation constant ($K_D$) for NADPH and TMP in binary and ternary complexes using Microscale Thermophoresis (MST), a powerful method for quantitative analysis of protein-ligand interactions (Table 5 and Supplemental Figs. S5–S7).

Despite its small size, TMP is a tight-binding inhibitor of EcDHFR[43,44]. Unsurprisingly, binding of TMP to EcDHFR yielded $K_D$ of 25 nM and a value in the picomolar range was anticipated in the presence of NADPH[45]. Accordingly, when we titrated TMP with a preincubated binary complex of EcDHFR:-NADPH, a highly cooperative binding event with an apparent $K_D$ of 11 pM was detected (>2000-fold difference). Next, we performed serial titration of NADPH with EcDHFR, and the apparent $K_D$ was determined to be 771 nM. In the presence of TMP, the $K_D$ for NADPH shifted ~100-fold to 8 nM, demonstrating a marked enhancement in cooperativity.

The potent affinity of TMP and impressive synergy with NADPH in EcDHFR is greatly diminished in DfrA1 and DfrA5. Analysis of TMP and co-factor binding individually to the unliganded form of both proteins resulted in $K_D$ values in the low micromolar range. When assessed in the bound state, the dissociation constant for NADPH was enhanced by merely 4- and 5-fold for DfrA1:TMP and DfrA5:TMP respectively, while the interactions of the inhibitor with the preformed binary DfrA1:NADPH and DfrA5:NADPH complexes demonstrated higher levels of cooperativity, improving 21- and 105-fold, respectively. A similar trend was reported with S. aureus DHFR and the TMP-resistant isoform, DHFR-S1, further reinforcing the notion that a diminished cooperativity of the inhibitor and NADPH is a hallmark of TMP resistance across many prokaryotic pathogens[46].

Finally, we probed if this attenuated interaction is also a factor in the binding specificity of UCP1228. Due to the low solubility of this compound, we limited our studies to the binding of NADPH to precomplexed DfrA1:UCP1228 and DfrA5:UCP1228 (Supplemental Fig. S8). A comparable enhancement in cooperativity of 4-fold as seen with DfrA1:TMP was found when NADPH was titrated into DfrA1:UCP1228. Interestingly, a more pronounced increase in binding synergy (15-fold) was detected when NADPH was added to preformed DfrA5:UCP1228 leading to the overall higher sensitivity of this enzyme to both TMP and PLAs, with $K_i$ values up to three times lower relative to DfrA1.

**Crystal structures of trimethoprim and propargyl-linked antifolates bound to EcDHFR, DfrA1, and DfrA5**. We determined ten crystal structures of EcDHFR, DfrA1 and DfrA5 in complex with TMP, UCP1223, and UCP1228 (Tables 1 and 2 and Supplementary Table S2). The X-ray structure of EcDHFR bound to TMP was first reported in 1982[47] and it became recently available through the Protein Data Bank (PDB:6XG5)[25]. In addition, we are disclosing for the first-time structures of DfrA1 and DfrA5 co-crystallized with TMP and the PLAs, UCP1223 and UCP1228. Together, these comprehensive crystallographic studies offer insight into the causes of diminished interactions of TMP,

**Table 2 Data collection and refinement statistics (molecular replacement).**

| | EcDHFR:UCP1223 | EcDHFR:UCP1228 | DfrA1:<br>NADPH:UCP1223 | DfrA1:<br>NADPH:UCP1228 | DfrA5:<br>NADPH:UCP1223 | DfrA5:<br>NADPH:UCP1228 |
|---|---|---|---|---|---|---|
| Beam line | SSRL 14-1 | SSRL 14-1 | SSRL 14-1 | UCONN COR$^2$E | SSRL 14-1 | SSRL 14-1 |
| PDB ID | 7REB | 7MQP | 7RGJ | 7REG | 7RGK | 7RGO |
| Space group | P6$_1$22 | P6$_1$22 | P3$_1$2 | P3$_1$2 | P4$_3$ | P4$_3$ |
| # monomers in ASU | 1 | 1 | 2 | 2 | 2 | 2 |
| Unit cell | | | | | | |
| $a, b, c$ (Å) | 66.49 66.49 213.19 | 66.68 66.68 213.75 | 72.28 72.28 120.15 | 71.86 71.86 119.87 | 99.42 99.42 43.40 | 98.85 98.85 43.14 |
| $\alpha, \beta, \gamma$ (°) | 90, 90, 120 | 90, 90, 120 | 90, 90, 120 | 90, 90, 120 | 90, 90, 90 | 90, 90, 90 |
| Resolution range (Å) | 34.27–1.91 (1.97–1.91) | 32.94–2.10 (2.17–2.10) | 30.29–1.44 (1.49–1.44) | 43.15–1.77 (1.83–1.77) | 31.44–2.19 (2.27–2.19) | 34.95–1.92 (1.98–1.92) |
| Completeness (%) | 97.73 (93.99) | 99.72 (100.00) | 99.27 (99.40) | 99.92 (99.94) | 99.84 (99.15) | 99.80 (99.69) |
| # unique reflections | 22,618 (2077) | 17,233 (1663) | 65,686 (6488) | 35,520 (3511) | 22,144 (2221) | 32,218 (3210) |
| I/$\sigma$ | 19.55 (2.15) | 21.82 (3.73) | 20.35 (3.46) | 15.37 (2.59) | 14.98 (2.41) | 9.87 (1.59) |
| $R_{work}$/$R_{free}$ | 0.226/0.248 | 0.200/0.241 | 0.198/0.227 | 0.190/0.220 | 0.184/0.229 | 0.192/0.227 |
| # non-H atoms | 1437 | 1439 | 2973 | 2815 | 2736 | 2803 |
| Protein | 1320 | 1317 | 2574 | 2502 | 2486 | 2503 |
| Ligands | 66 | 65 | 191 | 157 | 209 | 172 |
| Solvent | 51 | 57 | 208 | 156 | 41 | 128 |
| Average B-factor | 67.87 | 67.32 | 28.30 | 26.01 | 54.07 | 37.36 |
| Protein | 66.06 | 66.08 | 27.60 | 25.66 | 52.80 | 36.77 |
| Ligands | 104.93 | 91.79 | 29.64 | 27.09 | 68.49 | 41.74 |
| Solvent | 66.73 | 68.09 | 35.76 | 30.52 | 57.40 | 43.02 |
| RMS (bonds) (Å) | 0.008 | 0.020 | 0.006 | 0.007 | 0.008 | 0.008 |
| RMS (angles) (º) | 1.12 | 2.00 | 0.87 | 0.90 | 1.19 | 1.18 |
| Ramachandran | | | | | | |
| Favored (%) | 98.7 | 97.48 | 98.35 | 99.35 | 98.39 | 98.06 |
| Allowed (%) | 1.3 | 2.52 | 1.65 | 0.65 | 1.61 | 1.94 |
| Outliers (%) | 0 | 0 | 0 | 0 | 0 | 0 |
| Rotamer outliers (%) | 1.72 | 4.23 | 2.84 | 0.36 | 1.87 | 1.11 |

Values in parentheses are for highest-resolution shell.

mediated through the D27E/L28Q substitutions and reveal a common mechanism of resistance in TMP-insensitive DHFRs.

Expectedly, all three enzymes form the stereotypical DHFR fold with an eight-stranded β-sheet, four α-helices (B, C, E, and F) and three catalytic loops (the Met20, F-G, and G-H loops) (Fig. 3). Conditioned upon achieving proper orientation with co-factor, ligand binding is stabilized by electrostatic interactions between the protonated diaminopyrimidine ring (pKa = 7.4[48]) and the catalytically-required acidic residue at position 27, a well-established structural feature backed by biochemical and biophysical data[39,49–51]. However, structural differences in the Met20 loop implicate this region in a regulation of DHFR activity. Specifically, Asn18[EcDHFR] is replaced by Pro19[DfrA1/DfrA5] and Met20[EcDHFR] is replaced by Ile21[DfrA1/DfrA5] representing smaller, more rigid residues and presumably offering weaker interactions with inhibitors and modulating enzyme catalysis. Specifically, the substitution of Ile for Met20 would allow water to more readily access N5 of DHF. This water facilitates DHF protonation, and its entry to the active site is dictated by conformational changes of Met20[52–55]. Together with the hydrogen bonding capability of Gln29[DfrA1/DfrA5] these sequence variations likely affect the Met20 loop conformation and alter protein-substrate interactions.

Like DfrA1/DfrA5, human DHFR exhibits strong TMP insensitivity and it was interesting to probe for commonalities in these different reductases. One noteworthy similarity is that human DHFR, like DfrA5/DfrA1, also utilizes a glutamic acid (E30) to anchor folate substrates. The ternary DfrA5 structure was compared to an available structure (PDB ID: 2W3A) of human DHFR in complex with TMP and NADPH (Fig. 3). Surprisingly, TMP in the human enzyme overlays very closely with the conformation in DfrA5 including the twisted

arrangement of the trimethoxyphenyl ring. As with DfrA5/DfrA1, it appears that the E30 residue causes a displacement of TMP away from the co-factor binding site, thus eliminating many of the interactions necessary for strong binding. This observation raises the intriguing possibility that there are some similar structural themes that drive both TMP resistance in the plasmid-encoded enzymes and produce intrinsic insensitivity in the vertebrate enzyme[56].

**Loss of NADPH-trimethoprim cooperativity is evident in DfrA enzymes.** To better understand the impact of a less cooperative binding of NADPH in DfrA enzymes on TMP affinity and assess whether the D27E substitution disrupts NADPH-ligand interactions to impart resistance, we sought to crystallize ligands in both binary (DHFR:ligand) and ternary (DHFR:NADPH:ligand) complexes. Despite co-crystallization and soaking efforts, no density was observed for NADPH in the DfrA1:TMP structure, while in the DfrA5:TMP crystal structure the adenosine and phosphate moiety occupy their expected binding sites. Interestingly, the ribose moiety is oriented away from the active site and the electron density for the nicotinamide is not visible in the crystal structure presumably due to lack of interactions with the protein. This conformation is reminiscent of those observed in the occluded conformation of EcDHFR where Met16 directly blocks the binding of the nicotinamide ring[18]. However, structural alignment with both the occluded (PDB ID: 1RX6) and closed (PDB ID: 1RX2) EcDHFR complexes shows that the DfrA5 structure is much more closely related to the EcDHFR closed conformation. Therefore, this somewhat unique conformation of the DfrA5:TMP:NAPDH ternary complex likely

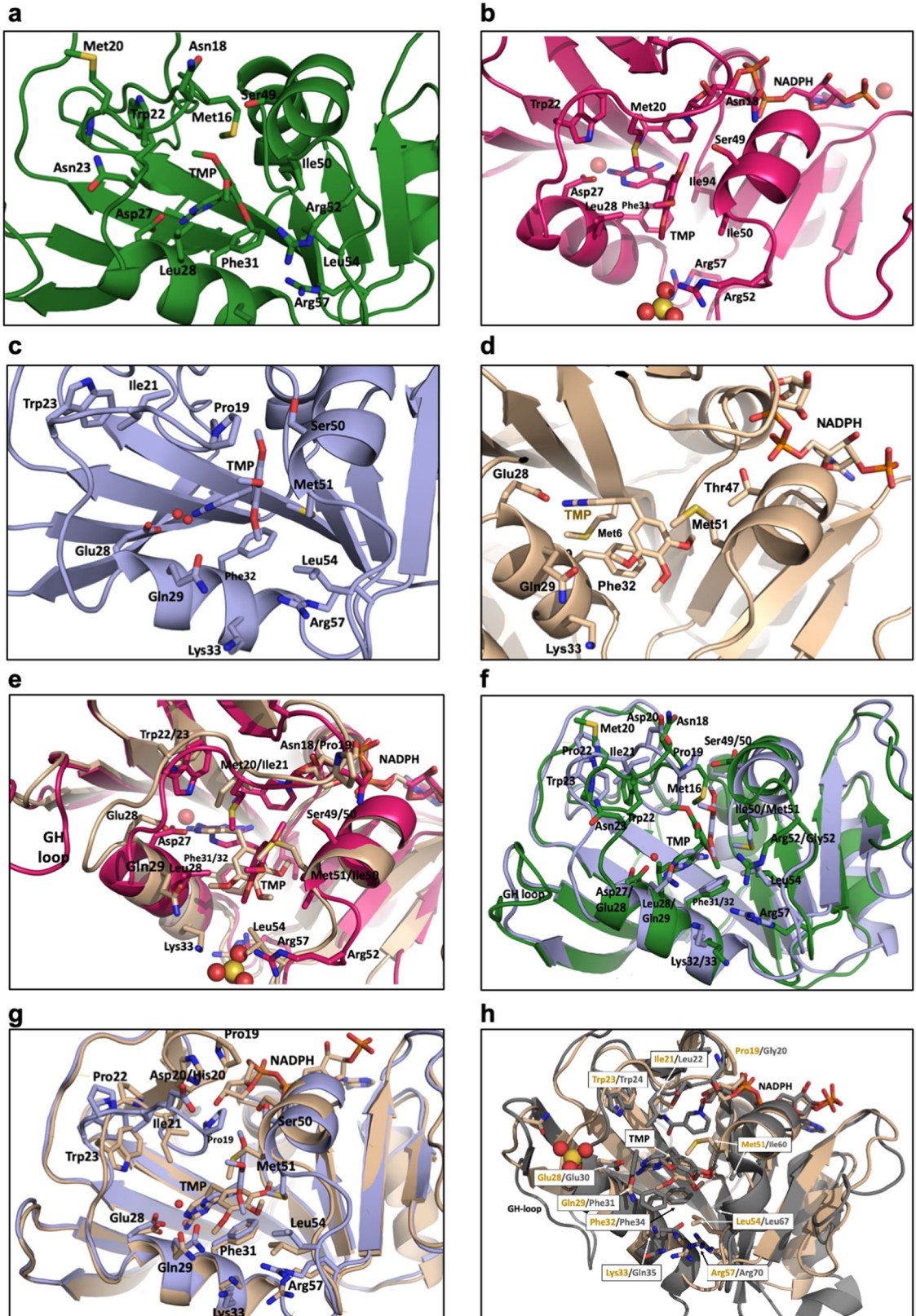

**Fig. 3 Crystal structures of DHFR with trimethoprim (TMP). a** EcDHFR:TMP (green), **b** EcDHFR:TMP:NADPH (magenta), **c** DfrA1:TMP (light blue), and **d** DfrA5:TMP:NADPH (beige). **e** Overlays of the crystal structures of DHFR with TMP ternary complexes; EcDHFR:TMP:NADPH (magenta) superimposed onto DfrA5:TMP:NADPH (beige), **f** Binary complexes; EcDHFR:TMP (green) superimposed onto DfrA1:TMP (light blue), **g** DfrA1:TMP (light blue) and DfrA5:TMP:NADPH (beige), and **h** DfrA5:TMP:NADPH (beige) superimposed onto Human DHFR:TMP:NADPH (PDB id code 2W3A). Residues important to ligand binding are shown in stick mode and colored by atom.

prevents favorable interactions between the nicotinamide ring and TMP and contributes to the reduced activity of TMP against this enzyme.

**D27E/L28Q substitutions result in alternative binding modes for trimethoprim and lead to antifolate resistance**. Crystal structures of *Ec*DHFR:TMP and *Ec*DHFR:NADPH:TMP, determined to 2.5 Å, and 3.0 Å respectively, show that the binding affinity of TMP is driven by electrostatic interactions between the protonated diaminopyrimidine ring and Asp27 as well as van der Waals contacts between the trimethoxyphenyl ring and helices B and C (specifically, the side chains of Leu28, Phe31, and Ile50) (Figs. 3 and S10). Furthermore, the antifolate adopts a nearly identical conformation when overlaid with chromosomal DHFRs from other prokaryotic species, including Gram-negative (*Coxiella burnetii*)[57] and Gram-positive (*Staphylococcus aureus*)[46] organisms.

The crystal structure of DfrA5:NADPH:TMP, solved to 2.6 Å, reveals major perturbations in the overall ligand binding pose compared to the structures of *Ec*DHFR (Figs. 3 and S10). Here, the diaminopyrimidine ring of TMP forms interactions with Glu28 (corresponding to Asp27 in EcDHFR); however, the trimethoxyphenyl moiety is oriented horizontally in the pocket, and key van der Waals contacts are lost through the L28Q substitution. Overlay of the *Ec*DHFR:NADPH:TMP and DfrA5:NADPH:TMP structures shows the D27E substitution lays the foundation for the divergent binding orientation of TMP (Fig. 3E). The additional methylene unit in the side chain of glutamic acid results in a 1.4 Å reorientation of the carboxylate head group, subsequently pulling TMP further into the active site by 1.1 Å. Variations in the GH loop likely enhance the positioning of TMP deeper in the active site of DfrA5 compared to *Ec*DHFR. Although distal to the active site, the GH loop makes critical contacts with helix B (residues 25–35) and is four residues shorter in DfrA5. Downstream effects ultimately cause a 1.9 Å shift in the Cα of Glu28[DfrA5] compared to Asp27[EcDHFR].

The crystal structure of DfrA1:TMP, determined to 2.2 Å, shows the antifolate interactions in the active site more closely resemble the binding mode observed in DfrA5 than in EcDHFR (Fig. 3). Similarly, electrostatic interactions between Glu28[DfrA1] and the diaminopyrimidine ring dictate the ligand binding mode. Like in DfrA5 crystal structures, the glutamic acid pulls TMP 0.5 Å further into the active site relative to the orientation of TMP in the *Ec*DHFR catalytic site, a phenomenon that is enhanced by a similarly shorter GH loop and a 2.0 Å shift in the Cα of E28[DfrA5] compared to D27[EcDHFR]. Additionally, critical hydrophobic contacts are lost through the L28Q substitution.

Excitingly, there are structural differences that rationalize the higher levels of TMP-resistance observed with DfrA1 compared to DfrA5. In DfrA1, a bridging water molecule facilitates the interaction between Glu28 and TMP, representing a loss of direct H-bond interactions. Another pronounced difference between the DfrA1:TMP and DfrA5:NADPH:TMP complexes is seen in the Met20 loop. In the DfrA1:TMP complex, this loop takes on a conformation reminiscent of the *Ec*DHFR occluded conformation with the Met20 loop occupying the NADPH binding site. The direct effect on TMP binding is through interactions with Met16[DfrA1] that result in a 2 Å push and ~90° rotation of the trimethoxyphenyl group as compared to the DfrA5:NADPH:TMP structure, correlating with a >3-fold increase in resistance between the two enzymes.

**Crystal structures of DfrA:PLAs reveal conformational flexibility that favors ternary complex formation and governs pan-DHFR inhibition**. Crystals of EcDHFR complexed with UCP1223 and UCP1228 diffracted to 2.1 Å and 1.9 Å, and

crystallized in the occluded conformation with the Met20 loop occupying the NADPH binding site (Figs. 4 and S11). Like TMP, the PLA binding mode is dictated through hydrogen bonding with Asp27[EcDHFR] or Glu28[DfrA1/DfrA5], and the propargyl linker forms π–π interactions with Phe31. However, unlike TMP, the bicyclic moiety forms van der Waals contacts with helix C (residues 43–50). These binding principles are maintained across all DfrA:PLA structures.

While UCP1223 and UCP1228 are structurally similar, their B-ring functionality differs substantially resulting in different orientations of the B- and C-rings in the DfrA active sites (Figs. 4 and S11 and S12). In the DfrA5:NADPH:UCP1223 structure, UCP1223 makes van der Waals contacts with Gln29, Met51, Leu54, and the bicyclic moiety is swung out of the pocket, exposing the highly polar benzylamine to bulk solvent. On the contrary, the DfrA1:NADPH:UCP1223 structure shows the benzylamine to form hydrogen bonds with the backbone carbonyls of residues Ile21 and Trp23, while the van der Waals contact with Met51 and Leu54 are more distant.

Comparatively, UCP1228 adopts a conformation that is in agreement with our previously published structures of PLAs co-crystallized with DHFR[28]. In both DfrA1 and DfrA5, the 2′-chloro group makes van der Waals contacts with surrounding residues Ile21, Thr47, Ser50, Met51, and the nicotinamide ring of NADPH (Fig. 4). This orients the biphenyl ring system such that it lays adjacent to helix C, picking up hydrophobic contacts with Met51 and Leu54. The binding pocket of the TMP-resistant enzymes is not capable of accommodating UCP1228 in the flipped conformer present in the UCP1223 structures while maintaining optimal interactions with the diaminopyrimidine pharmacophore. Interestingly, UCP1223 and UCP1228 have similar $K_i$ values across the DfrA enzymes, indicating that these substantially different conformational changes do not have a major effect on their binding potential.

Critically, unlike TMP, the PLAs preferentially form the ternary complex with NADPH in their respective DfrA complexes. When superimposed, both structures of DfrA1 and DfrA5 complexed with UCP1223 and UCP1228 show a clear density for NADPH. In comparing the DfrA1:NADPH:UCP1223 or UCP1228 structures with TMP, the Met20 loop is found in a closed conformation rather than an occluded conformation.

The PLA scaffold is able to accommodate the noted structural changes imparted by the D27E substitution allowing for conformational flexibility that favors ternary complex formation. Particularly, the propargyl linker replaces the methylene bridge of TMP extending the biphenyl ring system to better mimic the geometry of DHF. The linker likely promotes ternary complex stabilization through ligand-protein interactions, forcing the active site to adopt a conformation resembling the protein:cofactor:substrate complex, rather than through direct interactions with NADPH. This is apparent when comparing the scaffolds of UCP1223 and UCP1228 where only UCP1228 makes obvious contacts with NADPH through the 2′-chloro substitution. Inhibition data supports the notion that stabilizing the ternary complex of DHFR:NADPH:ligand is key to overcoming DfrA-mediated TMP resistance and developing pan-DHFR inhibitors.

## Conclusions

Collectively, these studies provided unique insight into the functional and structural features of the two most clinically relevant TMP-resistant DHFR isoforms, DfA1 and DfrA5, and are a critical first step in developing next-generation antifolates capable of overcoming widespread drug resistance. Both enzymes confer high levels of TMP resistance as demonstrated by over 1300- and 300-fold increases in $K_i$ compared to *Ec*DHFR. Through phylogenic studies and sequence analysis, we identified

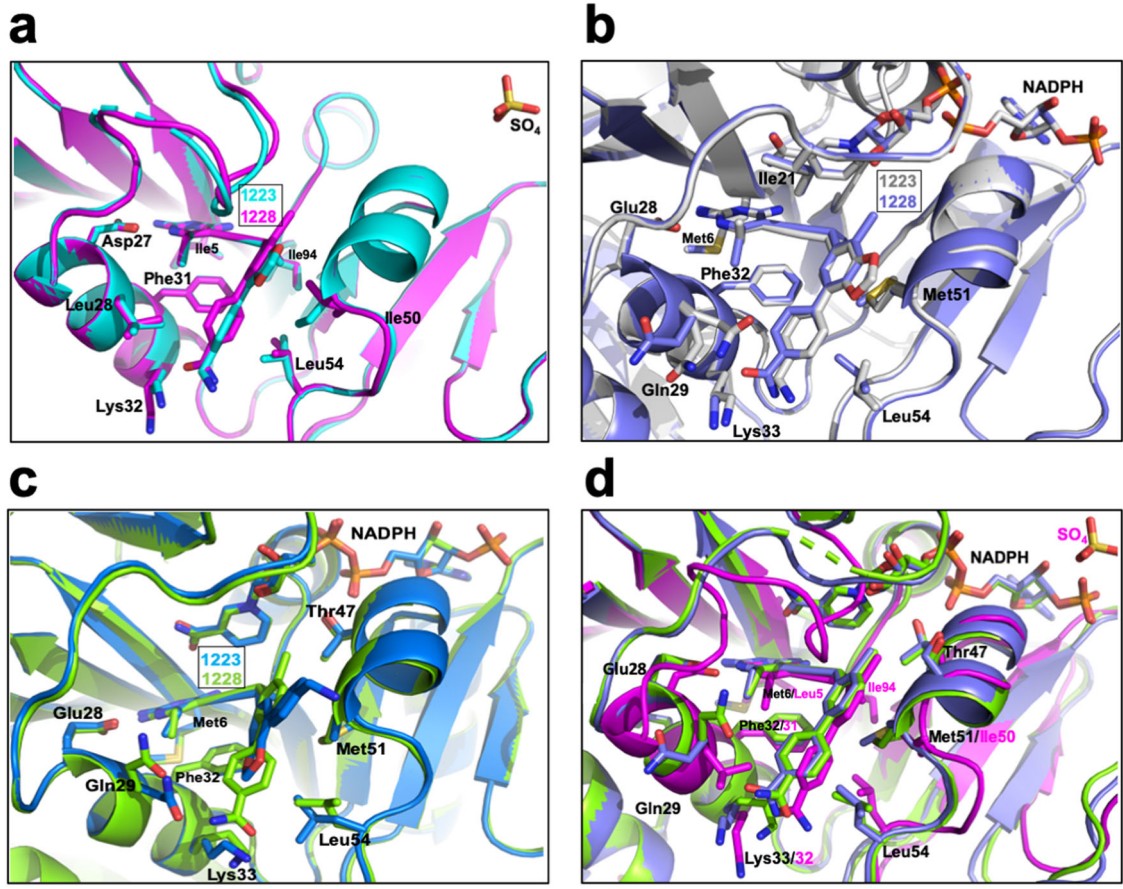

**Fig. 4 Superimposed crystal structures of DHFR with UCP1223 and UCP1228. a** Superposition of crystals structures of EcDHFR in complex with UCP1223 (cyan) and UCP1228 (magenta). **b** Superposition of crystals structures of DfrA1 in complex with UCP1223 (gray) and UCP1228 (purple). **c** Superposition of crystals structures of DfrA5 in complex with UCP1223 (blue) and UCP1228 (green). **d** Superposition of EcDHFR (magenta), DfrA1 (purple), and DfrA5 (green) in complex with UCP1228. Residues important to ligand binding are shown in stick mode and colored by atom.

D27E and L28Q substitutions as a common structural element in TMP-resistant DHFRs. Steady-state kinetic experiments linked these residue variations to a decreased affinity for the substrate while retaining enzyme function through enhancements in catalytic efficiency. While EcDHFR binds TMP almost 10-fold tighter than DHF, in the single point mutation variants, the affinity of substrate and inhibitor are almost similar. The presence of both mutations, D27E/L28Q ultimately tips the competitive landscape in favor of the substrate with a 3-fold decrease in affinity for the inhibitor. These observations support our structural data that shows the repositioning of the substrate, concomitant with the reduction of essential hydrophobic contacts within the substrate binding site, and owing to a much smaller size of TMP, loss of critical interactions with DfrA1 and DfrA5 enzymes as a basis for TMP resistance.

In addition, thermal unfolding data implicates NADPH as a major player in DHFR-ligand complex stabilization leading to the hypothesis that the loss of NADPH-ligand cooperativity and configuration change in cofactor is a hallmark of TMP resistance across different species[58] Pre-steady state experiments assessing the effects of NADPH on antifolate binding proved that this is a highly cooperative event, corresponding to a >2000-fold improvement in the TMP $K_d$ when NADPH is bound to EcDHFR. Large differences in the magnitude of NADPH/TMP cooperativity identified in DfrA1 and DfrA5 and the heavily biased selectivity of TMP toward bacterial DHFRs over vertebrate DHFR[42] correlate well with the enhanced presence of rigid and proline-rich substitutions in the TMP resistant enzymes that

likely prevent the active site Met20 loop from undergoing the large-scale conformational changes observed along the catalytic cycle of EcDHFR. The "allosteric wiring network" described by Chen et al[59]. referring to allosteric global conformational dynamics propagated upon ligand binding, is seemingly altered in the DfrA1 and DfrA5 forcing these enzymes into conformations where the Met20 loop sterically blocks NADPH access to the active site and results in reduced cooperativity with the inhibitor. Moreover, binding of TMP reorganizes DfrA1 into a conformation similar to the occluded conformation of EcDHFR that involves a notable shift of the Met20 loop. Of the ~500 structures of DHFR from various organisms available in the PDB, this is the first isoform of DHFR evolutionarily distinct from EcDHFR reported in the occluded state. To fully elucidate the functional relevance of conformational states observed here, in a rapid ligand exchange and co-factor facilitated product release events, other tools such as molecular dynamic simulations are being pursued[60]. Understanding allosteric features along the enzyme catalytic cycle will aid in design of next-generation pan-DHFR inhibitors with the special focus on the preferential formation and stabilization of the ternary complexes seen crystal structures of DfrA1 and DfrA5 bound to NADPH and PLAs.

## Methods

**Chemistry**. Trimethoprim was purchased from Sigma-Aldrich (Cat. No T7883). Methotrexate was purchased from Ark Pharm, Inc. (Cat. No. AK-77824). Iclaprim was a gift from Spero Therapeutics. Propargyl-linked antifolates evaluated in this work were synthesized through concise synthetic routes centered on key

palladium-coupling reactions[61–63]. Characterization of UCP1223 and UCP1228 by [1]H and [13]C nuclear magnetic resonance (NMR) spectra are presented in Supporting Material (Fig. S9).

**Cloning, expression, and purification of dihydrofolate reductase enzymes**. *E. coli* DHFR gene (UniProtKB:P0ABQ4) and *dfrA1* gene (UniProtKB:A4GRC7) were synthesized and cloned in pET-41a (+) vector with a C-terminal non-cleavable His tag via NdeI and XhoI restriction sites by GeneScript. *E. coli dfrA5* gene (UniProtKB:A0A4Y6L037) was synthesized and cloned in pET-24a (+) vector with the same restriction sites. All constructs were sequenced verified to ensure that no mutations were present. For the synthesis and sub-cloning of the *E. coli* DHFR mutants, a site-directed mutagenesis was performed to introduce single point mutations (Asp27 to Glu and Leu28 to Gln) as well as a double point mutation (Asp27 to Glu/Leu28 to Gln), using *E. coli* DHFR template previously synthesized by GeneScript.

Expression of these recombinant plasmids was followed by the same protocol. A single colony of BL21(DE3) (Novagen) transformed with the plasmid was incubated in 50 ml of LB media containing 30 μg/ml kanamycin, at 37 °C for 12–14 h with shaking at 225 rpm. The cell culture was diluted to 0.05 OD600 in freshly prepared LB with 30 μg/ml kanamycin and incubation continued until an OD600 reading of 0.7–0.8. The temperature was reduced to 30 °C and expression was induced with 1 mM isopropyl-β-D-thiogalactopyranoside (IPTG). After 6 h incubation the cells were harvested by centrifugation; the cell pellets were flash frozen in liquid nitrogen and stored at −80 °C. Cell pellets were thawed and resuspended with Lysis Buffer comprising 50 mM Tris pH 8.0, 0.4 M KCl, 5% glycerol, 5 mM Imidazole, 5 mM β-mercapthoethanol and 500 μg/ml lysozyme. One tablet of EDTA-free protease inhibitor (Roche Applied Science) was added per 10 ml of cell lysis buffer. After 20 min incubation on ice the cell suspension was disrupted by sonication (Qsonica 125 Ultrasonic Processor). Nucleic acids were then digested by adding 70 μg/ml of DNase I (Sigma). Cell debris was removed by centrifugation at 20 K × g for 25 min. The soluble lysate was filtered with a 0.45-μm filter and added to a nickel column (PerfectPro matrix) equilibrated with buffer A (25 mM Tris pH 8.0, 0.4 M KCl, 5 mM Imidazole, and 5 mM β-mercapthoethanol). Protein was eluted with buffer A supplemented with 250 mM Imidazole. Appropriate fractions identified by SDS-PAGE gel were pooled, concentrated, and desalted on PD10 disposable column (GE Healthcare) equilibrated with 20 mM Tris pH 8.0, 100 mM KCl, 15% glycerol, 0.1 mM EDTA, and 2 mM DTT. The eluted protein was aliquoted, flash frozen with liquid nitrogen and stored at −80 °C. The high level of protein purity was confirmed by SDS-PAGE stained with Coomassie Blue.

**Enzyme inhibition assay**. The assay was performed at room temperature using the Shimadzu UV/visible spectrophotometer. A standard reaction buffer contained 20 mM TES, pH 7.0, 50 mM KCl, 10 mM β-mercapthoethanol, 0.5 mM EDTA, and 1 mg/ml BSA. Protein solution (50 to 100 nM of purified recombinant DHFR) was mixed with 100 μM NADPH followed by the addition of test compound (dissolved in 100% DMSO solution), near the estimated IC50 value, to the mixture and incubated for 5 min. The enzymatic reaction was initiated with 100 μM DHF in 50 mM TES, pH 7.0. The rate of NADPH consumption during the conversion of dihydrofolate to THF was monitored by taking the absorbance reading at 340 nm, every 1 s over the initial 2 min. The differential extinction coefficient value of 13.2 mM[−1] cm[−1] (for the combination of conversion of NADPH to NADP + and DHF to THF) was used to convert the change in absorbance over time to an initial velocity. All measurements were performed in triplicate, from at least two independent experiments. Ki values reported in Table 3 were obtained using Cheng-Prusoff equation (Eq. 1)[64] to account for differing substrate affinity:

$$K_i = IC_{50}/(1 + [S]/K_M) \qquad (1)$$

**Antibacterial activity assessment**. Preparation, handling of cultures and the antibiotic susceptibility testing was performed according to CLSI guidelines. Briefly, *E. coli* BW25113 ATTC strain and *E. coli* JW0451 strain (Coli Genetic Stock Core) carrying a single acrB mutation were grown in Isosensitest Broth (ISO, Oxoid) overnight at 37 °C with shaking. In addition, the BL21(DE3) competent cells transformed with pET41a-*Ec*DHFR-6xHisTag, pET41a-DfrA1-6xHisTag, and pET24a-DfrA5-6xHisTag plasmids respectively were grown on selective media (LB + kan) overnight at 37 °C. On the day of the testing the organism was diluted to a final inoculum of 5 ×105 CFU/mL and 96-well round bottom plates (Corning INC., Corning NY) were prepared with 50 μl of ISO media. The compounds at the highest concentrations alone and paired with Sulfamethoxazole (1:19 w/w ratio) were then added to the first well before being serially diluted (2-fold) throughout the plate, leaving the last well with broth only (negative growth control). Next, 50 μl of the inoculum was added to each well. Plates were incubated at 37 °C for 18–20 h prior to reading. The minimal inhibitory concentration (MIC) defined as the lowest concentration of drug required to inhibit 80% of bacterial growth was assessed spectrophotometrically at Abs600, at the end time point. The MIC assay was run in duplicate and the results are reported in Supplemental Material Table S3.

**Steady-state kinetic measurements**. Dihydrofolate reductase was assayed spectrophotometrically for standard steady-state conditions at 25 °C. The kinetic parameters of Km DHF, Km NADPH and Ki DHF were obtained for both TMP-resistant DHFR enzymes and compared to the wild-type *E. coli* DHFR (EcDHFR) and its mutants. To measure the Km for DHF, a range of concentrations of the substrate (1 to 200 μM) was used while maintaining NADPH concentration constant. A similar approach was taken to obtain Km NADPH, where NADPH concentration was varied from 1 to 200 μM. The assay was initiated by adding DHF and monitoring NADPH oxidation at 340 nm. All measurements were performed in triplicate, from the same sample and are reported in Table 3. To obtain Km values, the initial velocity data was plotted as a function of DHF or NADPH concentration, analyzed by the Michaelis–Menten equation (Eq. 2):

$$Y = Vmax \times X/(Km + X) \qquad (2)$$

using a nonlinear regression method in GraphPad Prism 9.2.0 software. To determine the turnover number (kcat) using the same model, the enzyme concentration (based on the protein concentration determined by Bradford assay) was constrained to a constant value and analyzed by the following equation (Eq. 3):

$$Y = Et \times kcat \times X/(Km + X) \qquad (3)$$

The standard errors for Km and kcat were derived from a 95% confidence interval using Eqs. 4 and 5.

$$SD = \sqrt{(n)} \times (Upper\ limit - Lower\ limit)/(TINV(alpha, df \times 2)) \qquad (4)$$

where SD is the standard deviation, n is the sample size, alpha is 1–95% confidence interval (0.05), df is the degree of freedom (*n*−1).

$$SE = SD/\sqrt{(n)} \qquad (5)$$

where SE is the standard error, *n* is the sample size.

**Label-free differential scanning fluorimetry**. Thermal stability of *E. coli* DHFR mutants and the effect of ligand binding on thermal stability of *Ec*DHFR, DfrA1 and DfrA5 was analyzed by Tycho NT.6 (NanoTemper, Munich, Germany). In all, 200 μg of purified enzyme was preincubated for 18 h without and with a 10× molar excess of the ligand(s) (NADPH, TMP or PLA) in 20 mM Hepes 7.5, 100 mM KCl, 5% glycerol, ±2.0% DMSO and 2 mM DTT buffer. Thermal unfolding profiles were recorded within a temperature gradient (35 °C to 95 °C). Changes in intrinsic fluorescence intensity (measured ratio of absorbance at 350 nm over 330 nm) from tryptophan residues were used to calculate derivatives using the evaluation features provided by the Tycho instrument. The inflection temperature (Ti) as a marker of protein structural integrity was determined by the first derivative of the signal (ratio of 350 nm/330 nm). Mean value of Ti from at least three distinct samples with standard deviation are shown in Table S1 and Table 4. Representative unfolding profiles of each system, alone and in complex with the ligand are shown in Supplemental Material, Figs. S2–S4.

**Binding affinity analysis by microscale thermophoresis**. All measurements were carried out on a Monolith NT.115 device (NanoTemper Technologies GmbH), equipped with a RED and BLUE detection channel. High affinity RED-tris-NTA second-generation dye was used for labeling of His-tagged DHFR proteins (His6-*Ec*DHFR, His6-DfrA1 and His6-DfrA5) at 1:2 ratio of dye to protein to ensure a complete binding of the dye to the target. For the MST binding experiment of *Ec*DHFR with NADPH, in presence and absence of TMP, the protein and the dye were diluted in PBS buffer supplemented with 2% DMSO, 0.05% Tween-20, and ±1 μM TMP and labeled according to manufacturer's protocol. Same buffer was also used to prepare a 16-step serial dilution of NADPH, where the highest concentration of co-factor varied from μM to low nanomolar range, depending on the starting complex (see Supplemental Material Figs. S5–S8 for details). Labeled protein was then added to all dilutions for a final concentration of 10 or 50 nM enzyme, and after 1-hour incubation samples were loaded in Monolith NT.115 MST standard capillaries. The experiments were carried out using 40% and 60% MST power and between 40 and 80% LED power at 24 °C. The MST traces were recorded using standard parameters: 5 s MST power of, 30 s MST power on and 5 s MST power of. Titration of the non-fluorescent ligand for all 16 samples resulted in a gradual change in MST signal, which was plotted as ΔFnorm to yield a binding curve and was later fitted to derive the binding constants Kd using the MO.Affinity software provided by NanoTemper. For the high affinity binding determination of *Ec*DHFR with TMP, in presence and absence of 10 μM NADPH, the final concentration of the labeled protein and the titrant in the assay were adjusted to 10 nM and a low μM to picomole range, respectively. Same protocol was followed for the study of the interaction between Red-tris-NTA labeled DfrA1 and DfrA5 and the ligand (co-factor or TMP) in the presence and absence of the other binding partner. The labeling of the protein and the preparation of a serial dilution of the titrant were conducted in 20 mM Hepes pH 7.5, 100 mM KCl, 5% glycerol, 0.05 % Tween-20, and 2 mM DTT. The association was initiated by the addition of an equal volume of 100 nM DHFR enzyme labeled as an apo form or as a preformed binary complex with either 100 μM NADPH or 50 μM TMP to each reaction mixture containing 1.0 mM to 0.002 mM of the other ligand. Binding interactions between NADPH and the labeled binary complexes of DfrA1 and DfrA5 preincubated with 100 μM UCP1228 were characterized with the same parameters,

**Table 3 Summary of steady-state kinetic and inhibition constants of EcDHFR, DfrA1, and DfrA5.**

|  | EcDHFR | DfrA1 | DfrA5 | EcDHFR D27E | EcDHFR L28Q | EcDHFR D27E/ L28Q |
|---|---|---|---|---|---|---|
| **DHF steady-state kinetics** |  |  |  |  |  |  |
| $K_M$ (μM) | 3.2 ± 0.2 | 9.7 ± 2.1 | 24.2 ± 6.6 | 15.4 ± 2.1 | 5.2 ± 1.0 | 56.6 ± 8.0 |
| $k_{cat}$ (s$^{-1}$) | 6.7 ± 0.1 | 12.0 ± 0.6 | 31.4 ± 3.2 | 12.0 ± 0.6 | 5.1 ± 0.2 | 1.7 ± 0.09 |
| $k_{cat}/K_M$ (μM$^{-1}$*s$^{-1}$) | 2.1 ± 0.14 | 1.2 ± 0.3 | 1.3 ± 0.4 | 0.80 ± 0.1 | 0.99 ± 0.2 | 0.03 ± 0.004 |
| **NADPH steady-state kinetics** |  |  |  |  |  |  |
| $K_M$ (μM) | 5.8 ± 0.8 | 9.3 ± 1.7 | 19.9 ± 6.1 | 5.8 ± 0.9 | 7.6 ± 1.3 | 11.1 ± 1.6 |
| $k_{cat}$ (s$^{-1}$) | 3.2 ± 0.1 | 11.9 ± 0.6 | 34.3 ± 3.3 | 5.9 ± 0.3 | 4.4 ± 0.2 | 3.8 ± 0.2 |
| $k_{cat}/K_M$ (μM$^{-1}$*s$^{-1}$) | 0.6 ± 0.07 | 1.3 ± 0.2 | 1.7 ± 0.5 | 1.0 ± 0.2 | 0.6 ± 0.1 | 0.4 ± 0.05 |
| **Ki data (nM)[a]** |  |  |  |  |  |  |
| UCP1223 | 1.58 ± 0.26 | 20.33 ± 2.56 | 16.76 ± 0.29 | 12.75 ± 0.56 | 7.37 ± 0.72 | 187.14 ± 5.50 |
| UCP1228 | 2.73 ± 0.09 | 30.06 ± 3.35 | 20.08 ± 0.39 | 11.80 ± 0.40 | 10.53 ± 0.69 | 189.28 ± 5.36 |
| Trimethoprim | 0.43 ± 0.02 | 1332 ± 149 | 393.76 ± 81.8 | 7.25 ± 0.25 | 6.26 ± 0.30 | 171.07 ± 33.5 |
| Iclaprim | 1.64 ± 0.21 | 1143.7 ± 69 | 99.02 ± 13.9 | – | – | – |
| Methotrexate | 0.34 ± 0.05 | 177.36 ± 21 | 578.75 ± 73.8 | – | – | – |

Kinetic and inhibition parameters are based on the rate of conversion of NADPH and DHF to NADP$^+$ and THF, monitored by the changes in the absorbance reading at 340 nm. The kinetic constant, $K_M$ (μM) was generated by fitting initial velocity data as a function of DHF or NADPH concentration to the Michaelis–Menten model. The same model was used to determine the catalytic constant $k_{cat}$ (s$^{-1}$) where the enzyme concentration was constrained to a constant value. Average values from triplicates with standard error for each parameter are shown. The inhibition constants (Ki) were derived from Cheng-Prusoff equation (Eq. 1). The kinetic data demonstrate decreased affinity for DHF and NADPH in DfrA enzymes, concomitant with an increase in rate of catalysis resulting in similar catalytic efficiency. The inhibition data reveals a high level of resistance for clinical antifolates.
[a]Calculated as $K_i = IC_{50}/(1 + [S]/K_M)$ (Eq. 1).

**Table 4 Validation of different stabilizing effects of ligand binding on thermal stability of dihydrofolate reductases.**

**Clinically Relevant DHFR $T_i$ (°C)**

|  | EcDHFR | | | | DfrA1 | | DfrA5 | |
|---|---|---|---|---|---|---|---|---|
|  | $T_i$ 1 | $\Delta_1$ | $T_i$ 2 | $\Delta_2$ | $T_i$ | $\Delta$ | $T_i$ | $\Delta$ |
| Apo | 49.6 ± 0.6 | – | 60.2 ± 0.3 | – | 49.8 ± 0.5 | – | 66.3 ± 0.5 | – |
| +NADPH | 53.8 ± 0.4 | 4.2 | 63.1 ± 1.3 | 2.9 | 56.5 ± 0.8 | 6.7 | 69.2 ± 0.5 | 2.9 |
| +TMP | 67.1 ± 2.0 | 17.5 | 74.1 ± 0.5 | 13.9 | 52.3 ± 0.3 | 2.5 | 66.8 ± 0.7 | 0.5 |
| +NADPH, +TMP | – | – | 77.1 ± 0.5 | 16.9 | 62.3 ± 0.2 | 12.5 | 72.6 ± 0.3 | 6.3 |
| +UCP1223 | 68.5 ± 1.4 | 18.9 | 80.8 ± 0.2 | 20.6 | 58.5 ± 0.1 | 8.7 | 67.3 ± 1.0 | 1.0 |
| +NADPH, +UCP1223 | 78.9 ± 2.4 | 29.3 | 88.3 ± 0.5 | 28.1 | 71.2 ± 3.1 | 21.4 | 88.5 ± 0.2 | 22.2 |
| +UCP1228 | 65.5 ± 4.6 | 15.9 | 76.5 ± 2.5 | 16.3 | 55.4 ± 1.7 | 5.6 | 70. 0 ± 2.7 | 3.7 |
| +NADPH, +UCP1228 | 64.3 ± 0.1 | 14.7 | 75.7 ± 1.6 | 15.5 | 66.5 ± 3.3 | 16.7 | 77.3 ± 4.7 | 11 |

Thermal stability of EcDHFR, DfrA1 and DfrA5 enzyme as a function of co-factor and inhibitor binding was analyzed by Tycho NT.6 (NanoTemper, Munich, Germany). Temperature-dependent change in tryptophan fluorescence at emission wavelengths of 330 and 350 nm were used to calculate derivatives of the signal (ratio of 350 nm/330 nm). The maximum of the peak corresponds to the inflection point of the underlying ratio curve (inflection temperature, $T_i$). The gradual stabilizing effect upon ligand binding is reflected in the $T_i$ shift ($\Delta T_i$) between the apo form and the corresponding complex. Ti values, with their standard deviations are the averages from at least three independent measurements for each system.

whereas the dilution series of NADPH was ranging from high micromolar to low nanomolar concentrations. Supplementary Figures S5–S8 shows representative MST runs for each enzyme, pre and post incubation step, with equilibrium binding curves depicted as thick lines and dots representing up to 16 concentration sampling schemes. The line is the best fit of the data to the 1:1 binding model. Due to technical limitation of the Monolith NT.115 device, the very tight binding of TMP to the binary EcDHFR:NADPH complex, in a low picomolar range, resulted in the suboptimal binding curve with fewer points. Initial binding experiments for each studied system were conducted with a wide range of titrant concentration at medium MST power until the optimal binding conditions were achieved. Table 5 shows representative binding constants, $K_D \pm K_D$ confidence (±68% confidence) and the fold change in $K_D$ value ± relative error on the fold change, for each enzyme, pre and post incubation step, from the optimized MST experiments. The reported values are the sum of the fast, local environment-dependent responses of the fluorophore to the temperature jump and the slower diffusive thermophoresis fluorescence changes.

**Protein crystallography**. Crystallization experiments were performed by the hanging-drop vapor diffusion technique at 4 °C, using EasyXtal 15-well plates (Qiagen). Crystal Screen 1 and 2 (Hampton Research) were used to search for preliminary crystallization conditions. A typical sample was diluted to 5 mg/mL in 25 mM Tris pH 8.0, 100 mM KCl, 0.1 mM EDTA, 15% glycerol, and 1 mM DTT and incubated with 2.0 mM NADPH and 1.5 mM ligand (10 to 8-fold molar excess) for 18 h at 4 °C to ensure complex formation. Next day, the protein-ligand complex was concentrated to 25 mg/ml and mixed with a reservoir solution at 1:1

ratio. Crystals generally appeared within 1 to 2 weeks and were subsequently flash frozen in artificial mother liquid solution supplemented with 25% glycerol.

**Co-Crystallization of DfrA1 and DfrA5 with NADPH and Inhibitors**. Based on the already established crystallization conditions for DfrA1, pre-incubated complexes of DfrA1 and DfrA5 with co-factor and the ligand (UCP1223, UCP1228 or TMP) were plated with a set of PEG/CaCl$_2$ conditions. Within 5 days, X-ray quality crystals of DfrA1:NADPH:UCP1223 and DfrA1:NADPH:UCP1228 complex grew in reservoir solutions containing 15–19% PEG3350, 0.1 M imidazole pH 8.5. 0.3–0.35 M CaCl$_2$ and 2 mM DTT.

Optimization of the above conditions with different buffers and salts yielded well-defined crystal complexes of DfrA1:TMP and DfrA5:NADPH:PLA/TMP. DfrA1:TMP crystals grew from 22% PEG3350, 0.1 M NaCacodylate pH 6.5, 0.36 M MgCl$_2$, and 2 mM DTT. The ternary complexes of DfrA5:NADPH:UCP1223 and DfrA5:NADPH:UCP1228 crystallized in 13% to 19% PEG3350, 0.1 M MES pH 5.5–6.0, 0.3 M LiSO$_4$ and 2 mM DTT. DfrA5:NADPH:TMP crystals were obtained with 23% to 25% PEG3350, 0.1 M MES pH 6.0, 0.3 M LiSO$_4$ and 2 mM DTT.

**Co-Crystallization of wild-type E. coli DHFR (EcDHFR) with NADPH and Inhibitors**. Initial crystallization trials of the EcDHFR:NADPH:ligand samples resulted in small, needle-like crystals in conditions that were further optimized to produce large, hexagonal crystals suitable for diffraction. However, despite the use of an excess amount of NADPH in crystallization conditions only binary complexes of EcDHFR and ligand were observed. EcDHFR:UCP1223 and EcDHFR:UCP1228 complexes were crystallized in 10–13% PEG10K, 0.4–0.5 M

**Table 5 Binding affinity data of NADPH and TMP to EcDHFR, DfrA1 and DfrA5 by microscale thermophoresis (MST).**

| Complex | Titrant | $K_D$ (nM) | Fold change |
|---|---|---|---|
| **NADPH $K_D$ ± Trimethoprim** | | | |
| EcDHFR apo | NADPH | 771 ± 145 | – |
| EcDHFR:TMP | NADPH | 8.1 ± 4.0 | 95.2 ± 50.3 |
| DfrA1 apo | NADPH | 1650 ± 230 | – |
| DfrA1:TMP | NADPH | 430 ± 95 | 3.8 ± 1.0 |
| DfrA5 apo | NADPH | 7470 ± 1670 | – |
| DfrA5:TMP | NADPH | 1540 ± 380 | 4.9 ± 1.6 |
| **Trimethoprim $K_D$ ± NADPH** | | | |
| EcDHFR apo | TMP | 23.6 ± 5.1 | - |
| EcDHFR:NADPH | TMP | 0.011 ± 0.007 | 2145.5 ± 1441 |
| DfrA1 apo | TMP | 27,792 ± 4877 | – |
| DfrA1:NADPH | TMP | 1308 ± 451 | 21.2 ± 8.2 |
| DfrA5 apo | TMP | 11,929 ± 4448 | – |
| DfrA5:NADPH | TMP | 113 ± 56 | 105.6 ± 65.5 |
| **NADPH $K_D$ ± UCP1228** | | | |
| DfrA1 apo | NADPH | 1044 ± 106 | – |
| DfrA1:UCP1228 | NADPH | 289 ± 51 | 3.6 ± 0.7 |
| DfrA5 apo | NADPH | 6837 ± 1608 | – |
| DfrA5:UCP1228 | NADPH | 459 ± 157 | 14.9 ± 6.2 |

Binding response of the fluorescently labeled dihydrofolate reductases, pre and post incubation step to the increasing concentration of the non-fluorescent ligand. The Data were fitted to a single-site binding model accounting for ligand depletion. Binding constants, $K_D \pm K_D$ confidence (±68% confidence) and the fold change in $K_D$ value ± relative error on the fold change were determined from two independent measurements using MO Affinity Analysis v2.1.3 software (NanoTemper Technologies GmbH). Each binding experiment was prepared independently, in parallel using the same preparation of proteins.

ammonium sulfate and 2 mM DTT solution. Crystals of EcDHFR:TMP grew in 18% to 20% PEG4K or 8 K, 0.2 M ammonium sulfate and 2 mM DTT. Prior to cryoprotection and flash freezing, these crystals were transferred into the precipitant solution supplemented with 20% (v/v) glycerol and 10 mM NADPH for a short period of time (no longer that 5 min) that resulted in EcDHFR:NADPH:TMP complex.

**Data collection and refinement**. Diffraction data for EcDHFR:PLAs crystals were collected at the Stanford Synchrotron Radiation Lightsource (SSRL) on Beamline 14-1, and the data indexed and scaled using HKL 2000. PHASER was used for the initial molecular replacement, and the structures were refined and validated using non-crystallographic symmetry and structure restraints with the PHENIX software suite[65]. COOT[66] was used throughout the model building process. A model of EcDHFR (PDB ID: 1RF7) was initially used in the molecular replacement of EcDHFR bound to UCP1228, which was subsequently used to probe for the structure of EcDHFR bound to UCP1223. Diffraction data for EcDHFR with TMP, EcDHFR with NADPH/TMP and DfrA1 with TMP were collected at beam line 17-ID-1 (AMX) at NSLS-II. The data sets were processed in CCP4i2[67] using iMosflm[68] and molecular replacement was performed with MrBump[69] for EcDHFR with TMP, EcDHFR with NADPH/TMP and using PHASER[70] with PDB ID code 5ECC to solve DfrA1 with TMP. Rebuilding and refinement were done in CCP4i2 using COOT and Refmac5[71]. Diffraction data for DfrA1 bound to UCP1223 were collected at the Stanford Synchrotron Radiation Lightsource (SSRL) on Beamline 14-1, and the data indexed and scaled using HKL 2000. Diffraction data for DfrA1 bound to UCP1228 were collected at the University of Connecticut Center for Open Research Resource and Equipment (COR2E) on a Rigaku HighFlux HomeLab system, and d*TREK was used for indexing and scaling. For both structures, PHASER was used for the initial molecular replacement with a model of DfrA1 (PDB ID: 5ECC). The PHENIX software suite and COOT were used throughout the model building process and refinement. Diffraction data for DfrA5 bound to TMP and NADPH were collected at SSRL on Beamline 14-1. The PHENIX software suite and COOT were used throughout the model building process and refinement. The data sets were processed in CCP4i2 using iMosflm and molecular replacement PHASER with a model of previously solved DfrA5 bound to UCP1228. Data collection and refinement statistics are reported in Tables 1 and 2.

**Statistics and reproducibility**. Number of replicates is given for every experiment in the figure legends and the supplementary information. Results generally included a minimum of two biological replicates that are defined as the same or independent biological samples, performed in parallel or in subsequent experiments. The software GraphPad Prism (version 9.2.0) was used for kinetic analysis.

**Reporting summary**. Further information on research design is available in the Nature Research Reporting Summary linked to this article.

## Data availability

Data that supports the findings of this manuscript are deposited at the Protein Data Bank (PDB) with accession code 7REB, 7RGO, 7RGK, 7REG, 7RGJ, 7NAE, 7MYM, 7MYL, 7MQP, and 7R6G. All other data is available from the corresponding authors upon reasonable request.

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

## Acknowledgements

This work is supported by a generous grant from NIH/NIAID (1R01AI104841) to D.L.W. We would like to thank Stanford Synchrotron Radiation Light source (SSRL) and National Synchrotron Light Source II (NSLS-II) at Brookhaven National Lab for the beamline access and their staff members for their assistance. This research used resources of the National Synchrotron Light Source II, a U.S. Department of Energy (DOE) Office of Science User Facility operated for the DOE Office of Science by Brookhaven National Laboratory under Contract No. DE-SC0012704. The Center for BioMolecular Structure (CBMS) is primarily supported by the National Institutes of Health, National Institute of General Medical Sciences (NIGMS) through a Center Core P30 Grant (P30GM133893), and by the DOE Office of Biological and Environmental Research (KP1607011). Use of the Stanford Synchrotron Radiation Lightsource, SLAC National Accelerator Laboratory, is supported by the U.S. Department of Energy, Office of Science, Office of Basic Energy Sciences under Contract No. DE-AC02-76SF00515. The SSRL Structural Molecular Biology Program is supported by the DOE Office of Biological and Environmental Research, and by the National Institutes of Health, National Institute of General Medical Sciences (P30GM133894). We are grateful to the University of Connecticut Center for Open Research Sources and Equipment (COR$^2$E) for use of equipment and resources. The contents of this publication are solely the responsibility of the authors and do not necessarily represent the official views of NIGMS or NIH.

## Author contributions

J.K, M.N.L., and A.E. carried out the protein expression, purification, and crystallization. D.S. and K.V. synthesized and characterized UCP1223 and UCP1228. J.K. performed enzymes characterization and activity studies, thermostability, and thermophoretic analysis (Tycho and MST experiments). M.N.L., H.E, and A.E. carried out data collection, model building, refinement, and structural analysis. J.K., M.N.L., and D.L.W. wrote the manuscript with the contribution from other authors. All authors edited and approved the final version of the manuscript.

## Competing interests

The authors declare no competing interests.
