## [Peer Review File · Communications Biology]

Reviewers' comments:

Reviewer #1 (Remarks to the Author):

Recommendation: Major Revision

Comments:

Antimicrobial resistance is a growing public health crisis aggravated by the limited number of available treatment options, and the increasing rate of resistance to existing antibiotic drugs. While some scientists are investigating the detailed mechanisms of antibiotic resistance, others are in search of developing new drugs or trying to modify the existing drugs in order to improve their clinical use. The manuscript by Krucinska and co-workers describes the detailed kinetics and biophysical analysis of two *E. coli* DHFR isoforms (DfrA1 and DfrA5) and *E. coli* DHFR mutations (D27E, L28Q, and D27E/L28Q) in presence of clinically relevant DHFR inhibitors: trimethoprim, methotrexate, iclaprim, as well as, two of their propargyl-linked antifolates, UCP1223 and UCP1228. While both DfrA1 and DfrA5 enzymes confer high levels of TMP resistance compared to EcDHFR, the compounds UCP1223 and UCP1228 showed potential efficacy against these enzymes. The D27E/L28Q mutant significantly reduces the NADPH-ligand cooperativity leading to high TMP resistance. The authors have supported their data by label-free differential scanning fluorimetry and the binding affinity analysis by microscale thermophoresis. Several high-resolution binary and ternary co crystal structures are also provided to rationalize the change in conformation and binding affinities of these enzymes towards different inhibitors.

In their previous studies, the authors have showcased the activities of propargyl-linked antifolates against MRSA and *Streptococcus pyogenes* (PLoS ONE, 2012; Cell Chem. Biol. 2016), *Klebsiella pneumoniae* (Antimicrob. Agents Chemother. 2014), and *Mycobacterium tuberculosis* (PLoS ONE, 2016). This research is a thoughtful and logical extension to further explore the potential of these propargyl linked antifolates against *E. coli* and emphasizes solely on biochemical/structural characterizations. The manuscript is well-written and the data are presented in a systematic manner. Therefore, I recommend this manuscript for the publication in the Communications Biology after the authors address the following points:

Suggestions to the authors:

- "UCP1223 and UCP1228 maintain potent inhibitory activity against both TMP-resistant enzymes with K_i values ranging between 16-30 nM, only a 10-14-fold reduction relative to EcDHFR". Have the authors determined the MIC values of these compounds and compared them with TMP against "evolutionary related" *E. coli* with DfrA1, DfrA5 and *E. coli* having mutations: D27E, L28Q, and D27E/L28Q? Do the K_i values of these compounds determined herein reflect in the MIC values in whole cell assay? Can the "low solubility" of UCP1228 affect the cell-permeability of this compound and hence, its antibacterial property?

- The authors have previously demonstrated the activities of propargyl-linked antifolates against MRSA and *Streptococcus pyogenes* (PLoS ONE, 2012; Cell Chem. Biol. 2016), *Klebsiella pneumoniae* (Antimicrob. Agents Chemother. 2014), and *Mycobacterium tuberculosis* (PLoS ONE, 2016). What makes UCP1223 and UCP1228 more selective to *E. coli* DHFR, compared to other organisms?

- Page 9, line 213: "Although the X-ray structure of EcDHFR bound to TMP was reported in 1982 it is unavailable through the Protein Data Bank (PDB) and, to our knowledge, we are reporting the first contemporary structure for public use." A ternary crystal structure of EcDHFR:NADPH:TMP was recently reported in the literature (PDB: 6XG5).

- Fig. 3: The authors should include the electrostatic and van der Waal distances for every interactions showcased in all the panels.

- Figure S4: The authors must mention the solvent in which the NMR spectra were recorded.

Reviewer #2 (Remarks to the Author):

The manuscript describes a solid body of work comprising a range of different experimental techniques. Overall the work is important, interesting and timely, and the experimental work is certainly deserving of publication. There are however a number of issues in the discussion, and particularly with the scholarship of the work, that must be addressed before the manuscript is acceptable. These should be straightforward for the authors to deal with.

p1, line 18: PLA needs to be given in full here, not as the abbreviation.

p2, line 29: I presume this refers to design of carbapenemase inhibitors, not to the design of the enzymes themselves?

p2, lines 35-42: This paragraph would benefit from additional references for the first three sentences.

p2, line 42: This should presumably be "diminished by drug resistance".

p3, line 53: additional references would be useful for the first sentence of this paragraph.

p3, line 59: reference 13 would be better replaced by Sawaya and Kraut's 1997 Biochemistry paper describing the crystal structures of the various EcDHFR conformations, and Peter Wright's 2006 Science paper describing the NMR evidence.

p3, lines 63-64 and 74-75: The phylogenetic and sequence analysis mentioned here should be cited (if already published) or included as supplementary information (if not). Figure 1 is insufficient. Regardless, the idea that D27E mutations lead to trimethoprim resistance is not new; it is discussed in reference 5 of the present manuscript, and elsewhere. In fact an E28D mutation has been made in a TMP-resistant psychrophilic DHFR, and was shown not to affect TMP resistance (Loveridge et al, Protein Journal 2011, p546). Previous literature – not limited to that mentioned here – concerning the role of D27 in TMP resistance should be cited.

p4, line 80: a reference would be useful for the first sentence of this paragraph. The Sawaya and Kraut paper mentioned above would serve, although the original work was earlier than this.

p4, lines 81-82: The computational modelling mentioned here should be cited. Crystal structures of DHFRs with Glu at position 27 (or equivalent) are known (e.g. Hay et al, ChemBiochem 2009, p2348) and should also be cited.

p4, line 84: The use of the passive voice here makes it difficult to ascertain whether the prediction and hypothesis mentioned are the authors' own, or those of others. I suggest using the active voice to clarify this.

p4, line 86: To my knowledge, the D27E variant of EcDHFR was first generated and studied by Kraut's group 30 years ago (David et al, Biochemistry 1992, p9813). This should be cited and discussed in the manuscript. Some discussion of L28 variants would also be useful.

p4, line 92: This should presumably be "propargyl-linked antifolates".

p5, line 115: "reduction" should be "increase".

p6, lines 130-132: Hydride transfer in EcDHFR is ~20-fold faster than product release, so increases in the hydride transfer rate cannot possibly explain the elevated kcat values.

Section 2.4: It would be useful to see uncertainties on the inflection temperatures presented.

p6, line 144: "right shifted" would be better just as "increased".

p6, lines 172-174: This needs clarifying. DHFR inhibition cannot be driven by effects on hydride transfer or product release, as antifolate binding is competitive with substrate binding. If the inhibitor is bound, the substrate cannot bind. Hydride transfer therefore cannot take place, and no product is formed to be released. In any case I am not sure how the authors come to a conclusion about hydride transfer based on thermal stability data of inactive complexes.

p9, lines 221-223: The order of events is slightly out here. Evidence shows that dihydrofolate is not protonated when it binds EcDHFR, but that on binding the pKa is elevated from ~4 to ~6.5, allowing protonation from solvent water once bound. See (and cite), for example, Chen et al, *Biochemistry*, 1994, p7021, and Shrimpton et al, *Protein Science*, 2002, p1442, in addition to the Benkovic paper cited here.

p10, lines 229-230: Same issue as on p6 (lines 130-132) – changes to kcat are more likely to be related to physical steps of the catalytic cycle, not the chemical reaction itself.

p10, lines 237-242: This conformation of the nicotinamide ring of NADPH is well known; it was described by Sawaya and Kraut in the 1997 *Biochemistry* paper mentioned above, if not before. Various complexes of EcDHFR – both within the catalytic cycle and with inhibitors – show this “occluded conformation”; it is not anything to do with an “alternative mode of TMP” although it is related to the presence of certain folates (or antifolates). With NADP⁺ and folate bound, the closed conformation is adopted, but with NADP⁺ and THF bound the occluded conformation is adopted – a large change in conformation for a subtle change in ligand. In fact the authors mention the occluded conformation of EcDHFR on p11 and p12, so I am surprised that this link is not already made.

p10, line 247: At 2.5 Å the authors presumably cannot state whether the diaminopyrimidine ring is protonated or not. While protonation makes sense, in folates Asp27 interacts with the keto tautomer of the unprotonated pterin ring. Could the authors find (and cite) a pKa value for the diaminopyrimidine ring of TMP to support protonation?

p11: The discussion here focusses on comparison of DfrA1 and DfrA5 with EcDHFR itself. However, at least one structure of human DHFR with TMP bound is available through the PDB. As human DHFR does not bind TMP well, it would be useful to compare this structure to those obtained here.

p12, lines 284-285: The formation of the occluded conformation is unrelated to the concentration of NADPH used; it is related to structural changes imposed by the bound folate (or antifolate).

p13, lines 308-321: The discussion of the conformations is better here, but the authors should be careful on p317 as the “protein:substrate complex” is technically EcDHFR:DHF, which adopts an occluded conformation. The protein:cofactor:substrate complex adopts the closed conformation.

p14, lines 339-340: This discussion may need to be revised once my earlier comments have been addressed.

Materials and Methods: The last sentence of the “Chemistry” section needs re-writing. Use superscript numbers for atomic mass numbers.

Legend to Figure 1: Please cite the data mentioned.

Legend to Figure 2: The conversion is via hydride transfer from NADPH and protonation from solvent; please clarify.

Legend to Figure 4: The last sentence should be made consistent with equivalent sentences in the legend to Figure 3; in any case residues are not shown as ball-and-stick here (sticks only).

Table 1: Uncertainties are needed for all values given. Are Vmax values necessary when kcat is given? The legend would be better stating “based on the rate of conversion of NADPH and DHF to NADP⁺ and THF”, given the use of a combined extinction coefficient.

Table 2: Uncertainties are needed.

Table 3: Uncertainties are needed on the "fold change" values – these can be propagated from the errors on Kd (relative error on fold change = root of sum of squares of relative errors on Kd values).

Supplementary figures S1: please fix the x-axis to the bottom of these plots to make them look better

Figure S2A, II: presumably the green curve shows a "less pronounced" change?

Figures S4: Axes are missing from the spectra of UCP1228. Please state the solvents used. Use superscript numbers for atomic mass numbers in the legend (top of page).

Reviewer #1 (Remarks to the Author):

*(1) “UCP1223 and UCP1228 maintain potent inhibitory activity against both TMP-resistant enzymes with K_i values ranging between 16-30 nM, only a 10-14-fold reduction relative to EcDHFR”. Have the authors determined the MIC values of these compounds and compared them with TMP against “evolutionary related” *E. coli* with DfrA1, DfrA5 and *E. coli* having mutations: D27E, L28Q, and D27E/L28Q? Do the K_i values of these compounds determined herein reflect in the MIC values in whole cell assay? Can the “low solubility” of UCP1228 affect the cell-permeability of this compound and hence, its antibacterial property?*

We appreciate the reviewer’s concern as alterations to the chemical structure of an antibiotic can greatly impact the permeability into Gram-negative bacteria. This is in fact an active area of research in our program and we have additional upcoming manuscripts that probe these relationships in greater detail with a larger panel of inhibitors. Since the focus of the present manuscript was on the biophysical and structural aspects of antibiotic insensitivity in the TMP-resistant DHFR enzymes, we felt that these types of studies were beyond the scope of this already detailed paper. Nevertheless, we agree that it would be useful to the reader to demonstrate how the activity in a whole cell assay is impacted by the increase in size and hydrophobicity relative to TMP and have added new experiments and a new section to the manuscript (pg 5, ln 114-131) describing the activity of these two leads. We first assessed the antimicrobial activity and determined the MIC values of UCP1223 and UCP1228 alone and in combination with SMX (at 1:19 ratio) against a well-characterized standard *E. coli* strain (BW25113 from the Keio collection available through ATCC). We also evaluated the contribution of efflux machinery on PLA activity using an isogenic strain carrying a single ACR-B mutation (*E. coli* JW0451 strain). Interestingly, we found that the major factor limiting the activity of these PLAs is efflux rather than poor cellular permeability. This is consistent with other studies we are completing that shows that PLAs retain very good permeability into Gram-negative bacteria despite the increase in molecular weight and lipophilicity. These studies also showed that the powerful synergistic effects of combining DHFR inhibitors like TMP with SMX is also in operation with the PLA-based inhibitors.

Additionally, we went on to show that the compounds could maintain their antibacterial activity in the background of DfrA1 and DfrA5 enzyme. In this regard, we tested the susceptibility of *E. coli* BL21(DE3) cells transformed with pET41a-DfrA1 and pET24a-DfrA5 plasmids, respectively. These two strains exhibit high over-expression of the DfrA1 and DfrA5 protein under a T7 promoter, and show the expected reduced sensitivity relative to the parental strain. For comparative analysis, pET41a-EcDHFR plasmid, overexpressing EcDHFR wild type enzyme was also inserted into T7 expression strain and tested alongside the strains harboring the resistant elements. Combinations of UCP1223 and UCP1128 with SMX were still able to exert a measurable antibacterial effect on these overexpressing strains while TMP/SMX did not. Not only were we able to demonstrating that our compounds can still penetrate the Gram-negative bacteria but these studies provide additional target validation that the PLAs exert their antibacterial effects through DHFR inhibition. The preparation and handling of cultures and the antibiotic susceptibility testing method was performed according to CLSI guidelines. The corresponding data is discussed in the text, paragraph 2.2 and summarized in a new table added to the Supplemental Material (Table S4A-B).

(2) The authors have previously demonstrated the activities of propargyl-linked antifolates against MRSA and Streptococcus pyogenes (PLoS ONE, 2012; Cell Chem. Biol. 2016), Klebsiella pneumoniae (Antimicrob. Agents

Chemother. 2014), and *Mycobacterium tuberculosis* (PLoS ONE, 2016). What makes UCP1223 and UCP1228 more selective to *E. coli* DHFR, compared to other organisms?

This is an interesting point raised by the reviewer. We have typically used structure-based drug design to increase the potency of the inhibitors and have noted species-specific structural differences that can make certain compounds more active toward specific isoforms. However, there is a great deal of similarity in prokaryotic enzymes and we have identified several leads that are sufficiently active against enzymes from multiple bacterial/mycobacterial pathogens. We feel this is an important design element of our program aimed at broad-spectrum activity. As such, we have prioritized specific interactions that favor broad-spectrum inhibitory activity of the propargyl-linked antifolates against DHFR enzymes from both Gram-positive and Gram-negative pathogens. Towards that aim, we have successfully developed a large library of propargyl-linked antifolates including UCP1223 and UCP1228 compounds with an expanded spectrum efficacy against many chromosomal and plasmid-encoded TMP-resistant DHFR, not at all limited to *E. coli* organism.

(3) Page 9, line 213: “Although the X-ray structure of EcDHFR bound to TMP was reported in 1982 it is unavailable through the Protein Data Bank (PDB) and, to our knowledge, we are reporting the first contemporary structure for public use.” A ternary crystal structure of EcDHFR:NADPH:TMP was recently reported in the literature (PDB: 6XG5).

We thank the reviewer for pointing out this important omission. We were already in the process of preparing our manuscript when the ternary crystal structure of EcDHFR:NADPH:TMP complex became available (PDB: 6XG5). We have now properly cited and acknowledged it in the revised manuscript.

Manna MS, Tamer YT, Gaszek I, et al. A trimethoprim derivative impedes antibiotic resistance evolution. *Nat Commun.* 2021;12(1):2949. Published 2021 May 19. doi:10.1038/s41467-021-23191-z)

(4) Fig. 3: The authors should include the electrostatic and van der Waals distances for every interaction showcased in all the panels.

Thank you for the valuable suggestion. When we attempted to include the electrostatic and van der Waals distances for each interaction, Figure 3 became overly crowded and was very hard to evaluate. As an alternative, we created paired LigPlot+ figures for all ten crystal structures described in this study and have included them in the Supplemental Information, Figure S5A-C. This should allow the reader to easily access this important information.

(5) Figure S4: The authors must mention the solvent in which the NMR spectra were recorded.

Thank you for noting this omission. The appropriate NMR solvents have now been mentioned in the current spectra.

Reviewer #2 (Remarks to the Author):

(6) p1, line 18: PLA needs to be given in full here, not as the abbreviation.

We agree. The PLA abbreviation was replaced by a full name.

(7) p2, line 29: I presume this refers to design of carbapenemase inhibitors, not to the design of the enzymes themselves?

Yes, this sentence refers to inhibitors, not the enzymes and the error was corrected in the text.

(8) p2, lines 36-42: *This paragraph would benefit from additional references for the first three sentences.*

We appreciate the suggestion for additional citations. For page 2, line 36 the three new references cited below are now included:

Grim SA, Rapp RP, Martin CA, Evans ME. Trimethoprim-sulfamethoxazole as a viable treatment option for infections caused by methicillin-resistant *Staphylococcus aureus*. *Pharmacotherapy*. 2005 Feb;25(2):253-64. doi: 10.1592/phco.25.2.253.56956. PMID: 15767239.

Foster DR, Rhoney DH. *Enterobacter meningitis: organism susceptibilities, antimicrobial therapy and related outcomes*. *Surg Neurol*. 2005 Jun;63(6):533-7; discussion 537. doi: 10.1016/j.surneu.2004.06.018. PMID: 15936376.

Huovinen P. Resistance to trimethoprim-sulfamethoxazole. *Clin Infect Dis*. 2001 Jun 1;32(11):1608-14. doi: 10.1086/320532. Epub 2001 May 4. PMID: 11340533.

For page 2, line 38 the following citation was added:

Schnell JR, Dyson HJ, Wright PE. Structure, dynamics, and catalytic function of dihydrofolate reductase. *Annu Rev Biophys Biomol Struct*. 2004;33:119-40. doi: 10.1146/annurev.biophys.33.110502.133613. PMID: 15139807.

(9) p2, line 42: *This should presumably be “diminished by drug resistance”.*

Yes, this was a typo. The sentence in the text has been corrected.

(10) p3, line 53: *additional references would be useful for the first sentence of this paragraph.*

Two additional references have now been included on page 3, after line 56

Loveridge EJ, Allemann RK. Effect of pH on hydride transfer by *Escherichia coli* dihydrofolate reductase. *Chembiochem*. 2011 May 16;12(8):1258-62. doi: 10.1002/cbic.201000794. Epub 2011 Apr 19. PMID: 21506230.

Wan Q, Bennett BC, Wilson MA, Kovalevsky A, Langan P, Howell EE, Dealwis C. Toward resolving the catalytic mechanism of dihydrofolate reductase using neutron and ultrahigh-resolution X-ray crystallography. *Proc Natl Acad Sci U S A*. 2014 Dec 23;111(51):18225-30. doi: 10.1073/pnas.1415856111. Epub 2014 Dec 1. PMID: 25453083; PMCID: PMC4280638.
<https://doi.org/10.1073/pnas.1415856111>

(11) Page 3, line 59: *reference 13 would be better replaced by Sawaya and Kraut’s 1997 Biochemistry paper describing the crystal structures of the various EcDHFR conformations, and Peter Wright’s 2006 Science paper describing the NMR evidence.*

We thank the reviewer for highlighting these two important references which have now been substituted in place of reference 13:

Sawaya MR, Kraut J. Loop and subdomain movements in the mechanism of *Escherichia coli* dihydrofolate reductase: crystallographic evidence. *Biochemistry*. 1997 Jan 21;36(3):586-603. doi: 10.1021/bi962337c. PMID: 9012674.

Boehr DD, McElheny D, Dyson HJ, Wright PE. The dynamic energy landscape of dihydrofolate reductase catalysis. *Science*. 2006 Sep 15;313(5793):1638-42. doi: 10.1126/science.1130258. PMID: 16973882.

(12) p3, lines 63-64 and 74-75: *The phylogenetic and sequence analysis mentioned here should be cited (if already published) or included as supplementary information (if not). Figure 1 is insufficient. Regardless, the idea that D27E mutations lead to trimethoprim resistance is not new; it is discussed in reference 5 of the present manuscript, and elsewhere. In fact, an E28D mutation has been made in a TMP-resistant psychrophilic DHFR, and was shown not to affect TMP resistance (Loveridge et al, Protein Journal 2011, p546). Previous literature – not limited to that mentioned here – concerning the role of D27 in TMP resistance should be cited.*

We agree with the reviewer's assessment. We have not previously published this work so we have now included the phylogenetic and sequence analysis in the Supplementary Material, Figures S6A-B. Additionally, we have cited the following literature in the manuscript (page 3, line 65) in regard to prior discussions of the D27E mutation and its potential role in drug resistance.

Loveridge EJ, Dawson WM, Evans RM, Sobolewska A, Allemann RK. Reduced susceptibility of *Moritella profunda* dihydrofolate reductase to trimethoprim is not due to glutamate 28. *Protein J.* 2011 Dec;30(8):546-8. doi: 10.1007/s10930-011-9361-x. PMID: 21968646.

Manna MS, Tamer YT, Gaszek I, Poulides N, Ahmed A, Wang X, Toprak FCR, Woodard DR, Koh AY, Williams NS, Borek D, Atilgan AR, Hulleman JD, Atilgan C, Tambar U, Toprak E. A trimethoprim derivative impedes antibiotic resistance evolution. *Nat Commun.* 2021 May 19;12(1):2949. doi: 10.1038/s41467-021-23191-z. PMID: 34011959; PMCID: PMC8134463.

Howell EE, Villafranca JE, Warren MS, Oatley SJ, Kraut J. Functional role of aspartic acid-27 in dihydrofolate reductase revealed by mutagenesis. *Science.* 1986 Mar 7;231(4742):1123-8. doi: 10.1126/science.3511529. PMID: 3511529.

Appleman JR, Howell EE, Kraut J, Blakley RL. Role of aspartate 27 of dihydrofolate reductase from *Escherichia coli* in interconversion of active and inactive enzyme conformers and binding of NADPH. *J Biol Chem.* 1990 Apr 5;265(10):5579-84. PMID: 2108144.

(13) p4, line 80: *a reference would be useful for the first sentence of this paragraph. The Sawaya and Kraut paper mentioned above would serve, although the original work was earlier than this.*

As suggested by the reviewer we have cited the Sawaya and earlier work by Bystroff describing the interactions of diaminopyrimidine-based antifolates with the D27 residue.

Bystroff C, Kraut J. Crystal structure of unliganded *Escherichia coli* dihydrofolate reductase. Ligand-induced conformational changes and cooperativity in binding. *Biochemistry.* 1991 Feb 26;30(8):2227-39. doi: 10.1021/bi00222a028. PMID: 1998681.

(14) lines 81-82: *The computational modelling mentioned here should be cited. Crystal structures of DHFRs with Glu at position 27 (or equivalent) are known (e.g. Hay et al, ChemBiochem 2009, p2348) and should also be cited.*

We appreciate the reviewer's comment. We have added the suggested citations to the manuscript.

Hay S, et al. Are the catalytic properties of enzymes from piezophilic organisms pressure adapted? *ChemBioChem.* 2009;10(14):2348–23532.

Lombardo MN, G-Dayananandan N, Wright DL, Anderson AC. Crystal Structures of Trimethoprim-Resistant *DfrA1* Rationalize Potent Inhibition by Propargyl-Linked Antifolates. *ACS Infect Dis.* 2016 Feb 12;2(2):149-56. doi: 10.1021/acsinfecdis.5b00129. Epub 2016 Jan 4. PMID: 27624966; PMCID: PMC5108240.

(15) p4, line 84: *The use of the passive voice here makes it difficult to ascertain whether the prediction and hypothesis mentioned are the authors' own, or those of others. I suggest using the active voice to clarify this.*

(16) p4, line 86: *To my knowledge, the D27E variant of EcDHFR was first generated and studied by Kraut's group 30 years ago (David et al, Biochemistry 1992, p9813). This should be cited and discussed in the manuscript. Some discussion of L28 variants would also be useful.*

These are both excellent points. We have reworked the language in this paragraph to increase the clarity of the discussion and to add the appropriate citations. The revised section now reads as follows...

"In EcDHFR, Asp27 forms a critical electrostatic interaction with the basic headgroup of dihydrofolate-competitive inhibitors^{18,26}. Our analysis of the crystal structures of DfrA1 (PDB ID 5ECX) suggested that the resulting increase in side chain length due to the D27E substitution would reposition ligands, reducing TMP affinity while preserving substrate binding^{27,28}. In the case of L28Q substitution, Leu28 is proximal to hydrophobic regions of dihydrofolate and replacement by glutamine in DfrA proteins would likely alter substrate/inhibitor binding mode based on the same crystallographic evidence.

*Residue variations at Asp27 and Leu28 are known to have effects on EcDHFR catalysis. The D27E mutation was shown to increase ligand dissociation rates while maintaining the enzyme's ability to turnover substrate²⁹. This in turn might suggest that competitive inhibitors would be significantly impacted by the D27E substitution while the enzymatic activity would be preserved. Likewise, previous studies of DHFR variants demonstrated that mutations at EcDHFR's L28 have significant impact on enzyme efficiency and resistance to TMP³⁰. In addition, Wanger et al. found that the L28F mutation increased *k_{cat}* through interactions with the substrate/product³¹. We proceeded to introduce these substitutions in the chromosomal reductase to generate EcDHFR mutants harboring D27E, L28Q and D27E/L28Q substitutions as important comparators for our mechanistic studies on the DfrA1 and DfrA5 proteins."*

(17) p4, line 92: *This should presumably be "propargyl-linked antifolates".*

Yes, we corrected the typo.

(18) p5, line 115: *"reduction" should be "increase".*

This observation is correct. We have changed the "reduction" to "increase".

(19) p6, lines 130-132: *Hydride transfer in EcDHFR is ~20-fold faster than product release, so increases in the hydride transfer rate cannot possibly explain the elevated *k_{cat}* values.*

This is an excellent point raised by the reviewer. We have updated the text to read as follows:

"...presumably through increased product release."

(20) Section 2.4: *It would be useful to see uncertainties on the inflection temperatures presented.*

We agree with the reviewer's comment and addressed it in full by performing two additional replicates and updating the presented data.

(21) p6, line 144: *"right shifted" would be better just as "increased".*

Thank you for the suggestion. The wording was changed accordingly.

(22) p6, lines 172-174: *This needs clarifying. DHFR inhibition cannot be driven by effects on hydride transfer or product release, as antifolate binding is competitive with substrate binding. If the inhibitor is bound, the substrate cannot bind. Hydride transfer therefore cannot take place, and no product is formed to be released. In any case I am not sure how the authors come to a conclusion about hydride transfer based on thermal stability data of inactive complexes.*

We agree with the reviewer that text was poorly crafted and confusing. We clarified our conclusions in the text by removing discussion of the reaction steps and including additional citations:

Original text:

“This data supports the hypothesis that DHFR inhibition is predominantly driven by effects on hydride transfer from NADPH to DHF and product release, the rate-limiting steps in the DHFR catalytic cycle, rather than outcompeting substrate binding.”

Changed to:

“This data supports the idea that DHFR inhibition is strongly influenced by the relative positioning of the antifolate and the NADPH cofactor, such that an increase in cofactor:ligand:enzyme interactions significantly stabilize the ternary complex.”

(23) p9, lines 221-223: *The order of events is slightly out here. Evidence shows that dihydrofolate is not protonated when it binds EcDHFR, but that on binding the pKa is elevated from ~4 to ~6.5, allowing protonation from solvent water once bound. See (and cite), for example, Chen et al, Biochemistry, 1994, p7021, and Shrimpton et al, Protein Science, 2002, p1442, in addition to the Benkovic paper cited here.*

We agree with the reviewer on the order of binding events. In this section, we are simply describing the hydrogen bonds required for ligand binding and not the events leading to binding. We have adjusted the text to be more clear on this point.

Original text:

“Ligand binding is driven by electrostatic interactions between the protonated diaminopyrimidine ring and the catalytically-required acidic residue at position 27, a well-established structural feature backed by biochemical and biophysical data.”

Revised text:

“Conditioned upon achieving proper orientation with cofactor, ligand binding is stabilized by electrostatic interactions between the protonated diaminopyrimidine ring (pKa = 7.4⁴⁸) and the catalytically-required acidic residue at position 27, a well-established structural feature backed by biochemical and biophysical data^{39,49-51}”

...and cited the two additional papers as suggested by the reviewer:

Chen YQ, Kraut J, Blakley RL, Callender R. Determination by Raman spectroscopy of the pKa of N5 of dihydrofolate bound to dihydrofolate reductase: mechanistic implications. Biochemistry. 1994 Jun 14;33(23):7021-6. doi: 10.1021/bi00189a001. PMID: 8003467.

Shrimpton P, Allemann RK. Role of water in the catalytic cycle of E. coli dihydrofolate reductase. Protein Sci. 2002 Jun;11(6):1442-51. doi: 10.1110/ps.5060102. PMID: 12021443; PMCID: PMC2373639.

(24) p10, lines 229-230: *Same issue as on p6 (lines 130-132) – changes to kcat are more likely to be related to physical steps of the catalytic cycle, not the chemical reaction itself.*

We appreciate the comment. We agree with the reviewer and we altered the text to remove reference to chemical reaction.

Original text:

“...these alterations likely attribute to the observed 10-fold enhancement in the catalytic rate constant relative to EcDHFR.”

Changed to:

“...these sequence variations likely affect the Met20 loop conformation and alter protein-substrate interactions.”

(25) p10, lines 237-242: This conformation of the nicotinamide ring of NADPH is well known; it was described by Sawaya and Kraut in the 1997 Biochemistry paper mentioned above, if not before. Various complexes of EcDHFR – both within the catalytic cycle and with inhibitors – show this “occluded conformation”; it is not anything to do with an “alternative mode of TMP” although it is related to the presence of certain folates (or antifolates). With NADP⁺ and folate bound, the closed conformation is adopted, but with NADP⁺ and THF bound the occluded conformation is adopted – a large change in conformation for a subtle change in ligand. In fact the authors mention the occluded conformation of EcDHFR on p11 and p12, so I am surprised that this link is not already made.

While the conformation of NADPH we discuss here is similar to that observed in the occluded EcDHFR structures, this ternary complex is more related to the closed form. The crystal structure of DfrA5 bound with TMP and NADPH shows the enzyme adopts a closed conformation. When DfrA5:NADPH:TMP is aligned with the closed structure reported by Sawaya et al (PDB ID: 1RX2), we see the C-alpha of Cys17 in DfrA5 aligns with the C-alpha of Met16 in *E. coli* with a distance of 0.6Å (alignment RMSD: 0.806). This is contrary to the occluded conformation where Met16 occupies the NADPH binding site. In the alignment of DfrA5:NADPH:TMP with the occluded structure reported by Sawaya et al (PDB ID: 1RX6), the distance between the C-alpha of Cys17 in DfrA5 and the C-alpha of Met16 is 6.6Å (alignment RMSD: 0.897). To our knowledge, TMP has been only reported to bind prokaryotic enzymes such that the trimethoxy ring is oriented vertically in the active site and this is the first reported alternative binding mode of TMP (horizontally in the pocket) in a bacterial species. A consequence of this different binding mode is the inability of DfrA5 to bind both TMP and the nicotinamide ring of NADPH which becomes a driving factor in resistance. We have revised the text as given below to better clarify this result in the manuscript. we have added this statement to the text:

“Interestingly, the ribose moiety is oriented away from the active site and the electron density for the nicotinamide is not visible in the crystal structure presumably due to lack of interactions with the protein. This conformation is reminiscent of those observed in the occluded conformation of EcDHFR where Met16 directly blocks the binding of the nicotinamide ring. [Sawaya MR, Kraut J. Loop and subdomain movements in the mechanism of Escherichia coli dihydrofolate reductase: crystallographic evidence. Biochemistry. 1997 Jan 21;36(3):586-603. doi: 10.1021/bi962337c. PMID: 9012674]. However, structural alignment with both the occluded (PDB ID: 1RX6) and closed (PDB ID: 1RX2) EcDHFR complexes shows that the DfrA5 structure is much more closely related to the EcDHFR closed conformation. Therefore, this somewhat unique conformation of the DfrA5:TMP:NADPH ternary complex likely prevents favorable interactions between the nicotinamide ring and TMP and contributes to the reduced activity of TMP against this enzyme.”

(26) p10, line 247: At 2.5 Å the authors presumably cannot state whether the diaminopyrimidine ring is protonated or not. While protonation makes sense, in folates Asp27 interacts with the keto tautomer of the unprotonated pterin ring. Could the authors find (and cite) a pKa value for the diaminopyrimidine ring of TMP to support protonation?

We have added the pKa value (7.4) to the text and the following citation:

Aagaard, J.; Madsen, P. O.; Rhodes, P.; Gasser, T. MICs of ciprofloxacin and trimethoprim for *Escherichia coli*: Influence of pH, inoculum size and various body fluids. *Infection* **19**, (1991). [PMID:2055655]

(27) p11: *The discussion here focusses on comparison of DfrA1 and DfrA5 with EcDHFR itself. However, at least one structure of human DHFR with TMP bound is available through the PDB. As human DHFR does not bind TMP well, it would be useful to compare this structure to those obtained here.*

We thank the reviewer for this excellent suggestion. We have added a new panel (H) to Figure 3 which shows ternary structure of DfrA5 in complex with TMP and NADPH overlaid with the structure of human DHFR bound to TMP and NADPH (PDB:2W3A). Interestingly, we found that the TMP pose observed in DfrA1/DfrA5 is very similar to that seen in human and may suggest some similarities in the structural basis of TMP resistance between the plasmid-borne and human enzymes. A short comparative analysis has been added to the paragraph 2.6 (pg. 11, ln 260-268) to provide a clear illustration of major structural differences between the structures and reads as follows:

“Like DfrA1/DfrA5, human DHFR exhibits strong TMP insensitivity and it was interesting to probe for commonalities in these different reductases. One noteworthy similarity is that human DHFR, like DfrA5/DfrA1, also utilizes a glutamic acid (E30) to anchor folate substrates. The ternary DfrA5 structure was compared to an available structure (PDB ID: 2W3A) of human DHFR in complex with TMP and NADPH (Figure 3H). Surprisingly, TMP in the human enzyme overlays very closely with the conformation in DfrA5 including the twisted arrangement of the trimethoxyphenyl ring. As with DfrA5/DfrA1, it appears that the E30 residue causes a displacement of TMP away from the co-factor binding site, thus eliminating many of the interactions necessary for strong binding. This observation raises the intriguing possibility that there are some similar structural themes that drive both TMP resistance in the plasmid-encoded enzymes and produce intrinsic insensitivity in the vertebrate enzyme⁵⁶”

(28) p12, lines 284-285: *The formation of the occluded conformation is unrelated to the concentration of NADPH used; it is related to structural changes imposed by the bound folate (or antifolate).*

Thank you for pointing it out. We recognize that the conformation of *E. coli* DHFR is dictated by the substrate/ligand rather than cofactor and have changed the text to remove any suggestion otherwise.

“Crystals of EcDHFR complexed with UCP1223 and UCP1228 diffracted to 2.1Å and 1.9Å, and crystallized in the occluded conformation with the Met20 loop occupying the NADPH binding site (Figure 4A).”

(29) p13, lines 308-321: *The discussion of the conformations is better here, but the authors should be careful on p317 as the “protein:substrate complex” is technically EcDHFR:DHFR, which adopts an occluded conformation. The protein:cofactor:substrate complex adopts the closed conformation.*

This is a good note by the reviewer. We have revised the text to be more specific and hope that it is now clearer. New text reads as follows:

“Particularly, the propargyl linker replaces the methylene bridge of TMP extending the biphenyl ring system to better mimic the geometry of DHF. The linker likely promotes ternary complex stabilization through ligand-protein interactions, forcing the active site to adopt a conformation resembling the protein:cofactor:substrate complex, rather than through direct interactions with NADPH.”

(30) p14, lines 339-340: *This discussion may need to be revised once my earlier comments have been addressed.*

We appreciate the thoughtful comments of the reviewer and feel that they have improved the quality of the discussion throughout the text. While our main conclusions remain largely the same, we have updated the conclusions paragraph to include the following observations.

Page 15, line 362:

“While EcDHFR binds TMP almost 10-fold tighter than DHF, in the single point mutation variants, the affinity of substrate and inhibitor are almost similar. The presence of both mutations, D27E/L28Q ultimately tips the competitive landscape in favor of the substrate with a 3-fold decrease in affinity for the inhibitor. These observations support our structural data that shows the repositioning of the substrate, concomitant with the reduction of essential hydrophobic contacts within the substrate binding site and owing to a much smaller size of TMP, loss of critical interactions with DfrA1 and DfrA5 enzymes, as a basis for TMP resistance.”

Page 15, line 373:

“Significant differences in the magnitude of NADPH/TMP cooperativity identified in DfrA1 and DfrA5 and the heavily biased selectivity of TMP toward bacterial DHFRs over vertebrate DHFR (Ref. Baccanari DP, Kuyper LF. Basis of selectivity of antibacterial diaminopyrimidines. J Chemother. 1993 Dec;5(6):393-9. PMID: 8195830) correlate well with the enhanced presence of rigid and proline-rich substitutions in the TMP resistant enzymes that likely prevent the active site Met20 loop from undergoing the large-scale conformational changes observed along the catalytic cycle of EcDHFR.”

(31) Materials and Methods: The last sentence of the “Chemistry” section needs re-writing. Use superscript numbers for atomic mass numbers.

We appreciate the comment. This has been fixed.

(32) Legend to Figure 1: Please cite the data mentioned.

The sentence in Figure 1 was rephrased to be more concise.

(33) Legend to Figure 2: The conversion is via hydride transfer from NADPH and protonation from solvent; please clarify.

We apologize for our lack of clarity. We have now incorporated the suggestion in the legend to Figure 2.

(34) Legend to Figure 4: The last sentence should be made consistent with equivalent sentences in the legend to Figure 3; in any case residues are not shown as ball-and-stick here (sticks only).

We have corrected the legend to Figure 4.

(35) Table 1: Uncertainties are needed for all values given. Are Vmax values necessary when kcat is given? The legend would be better stating “based on the rate of conversion of NADPH and DHF to NADP+ and THF”, given the use of a combined extinction coefficient.

Thank you for the observation. We have incorporated the suggested contents to the Table 1 and the legend respectively.

(36) Table 2: Uncertainties are needed.

Based on the comment of the reviewer we have performed additional Tycho experiments in order to obtain standard errors for all the reported Ti values in Table 2.

(37) Table 3: Uncertainties are needed on the “fold change” values – these can be propagated from the errors on Kd (relative error on fold change = root of sum of squares of relative errors on Kd values).

Thank you for pointing it out. The uncertainties on the fold change values in Table 3 were calculated as suggested by the reviewer and reported accordingly.

(38) Supplementary figures S1: please fix the x-axis to the bottom of these plots to make them look better

We appreciate the reviewer's observation; the problem has been fixed.

(39) Figure S2A, II: presumably the green curve shows a “less pronounced” change?

Yes, the statement describing the thermophoretic change was corrected to a "less pronounced" to reflect the smaller degree of response (amplitude) upon NADPH binding event.

(40) Figures S4: Axes are missing from the spectra of UCP1228. Please state the solvents used. Use superscript numbers for atomic mass numbers in the legend (top of page).

Thank you for identifying this issue. The spectra of UCP1228 have been replaced by a new one with proper axes. In each spectra respective NMR solvents have been mentioned. Superscripts were used to represent atomic mass numbers in the caption.

Sincerely,

Dennis L. Wright
Professor of Pharmaceutical Sciences
And Chemistry

REVIEWERS' COMMENTS:

Reviewer #1 (Remarks to the Author):

The authors have addressed most of the points raised by the reviewers and made the necessary changes in the manuscript. I, therefore, recommend this manuscript for the publication in the *Communications Biology* in its current form.

Reviewer #3 (Remarks to the Author):

The authors have provided a revised manuscript which addresses all my comments of substance, although a few trivial points remain (see below). As indicated in my previous review, the work is important, interesting and timely, and the experimental work is certainly deserving of publication. The revised manuscript addresses all reservations I had about the work, and I am pleased to recommend publication. I look forward to seeing the manuscript in print.

Trivial points (can be addressed now or in proof):

p4, line 102: the authors have corrected their previous typo by removing the phrase "propargyl-linked antifolates" entirely and leaving only the abbreviation (in brackets). As this is the first time the abbreviation is used in the main body of the manuscript text, I suggest writing it in full here.

In the "Chemistry" section of the Materials and Methods, atomic mass numbers and reference numbers all need to be superscript.

The legend to Figure 2 does not appear to have been significantly changed; my original point is that protonation from solvent must also be mentioned, in addition to hydride transfer from NADPH. I apologise for the lack of clarity in my request for clarity.

Reviewer #3 (Remarks to the Author):

p4, line 102: the authors have corrected their previous typo by removing the phrase “propargyl-linked antifolates” entirely and leaving only the abbreviation (in brackets). As this is the first time the abbreviation is used in the main body of the manuscript text, I suggest writing it in full here.

This has been changed as suggested

In the “Chemistry” section of the Materials and Methods, atomic mass numbers and reference numbers all need to be superscript.

These have been corrected

The legend to Figure 2 does not appear to have been significantly changed; my original point is that protonation from solvent must also be mentioned, in addition to hydride transfer from NADPH. I apologise for the lack of clarity in my request for clarity.

This has been added to the figure legend

Sincerely,

Dennis L. Wright
Professor of Pharmaceutical Sciences
And Chemistry